# GRIT: Geometry-Aware PEFT with K-FAC Preconditioning, Fisher-Guided Reprojection, and Dynamic Rank Adaptation

## Abstract

**Parameter-efficient fine-tuning (PEFT)** is now the standard approach for adapting LLMs to *specific domains and use cases*, yet prominent approaches such as *LoRA* and *QLoRA* are largely *geometry-agnostic*: they optimize within *fixed, randomly oriented low-rank subspaces* using plain first-order descent, *ignoring* local **loss curvature**. This *inflates the parameter–update budget* and *increases drift* along weakly constrained directions. We introduce **GRIT**, which turns standard LoRA updates into a *dynamic, curvature-aware* procedure. Concretely, GRIT retains the LoRA parameterization but: (1) **preconditions gradients** in the adapter's rank space using *K-FAC (Kronecker-Factored Approximate Curvature)* as a natural-gradient proxy; (2) periodically **reprojects** the low-rank basis onto dominant *Fisher eigendirections* to suppress drift; and (3) **adapts the effective rank** by reading the spectrum so capacity concentrates where the signal is. The overall effect is to steer updates into *high-signal, low-interference* directions while using *fewer effective parameters*. Across *instruction-following*, *comprehension*, and *reasoning* benchmarks on LLaMA backbones, **GRIT** *matches or surpasses LoRA/QLoRA* while **cutting trainable parameters by** $\sim 46\%$ **on average** (*25–80% across tasks*) *without degrading quality*. **Fine-tuning large language models** typically induces *catastrophic forgetting*—drift from the pretraining distribution that erodes general knowledge. We model GRIT's forgetting with a curvature-modulated power law $\boxed{L_{pt}^{\mathrm{GRIT}} = L_{pt}^0 + A \dfrac{D_{ft}^{\beta}}{(\Xi_{\mathrm{GRIT}} N)^{\alpha}} + E}$ where, in compact form, $\Xi_{\mathrm{GRIT}} = (1 + \gamma_r r_{\mathrm{eff}})(1 + \gamma_a \rho_{\mathrm{align}})(1 + \gamma_p \pi_{\mathrm{proj}})$ captures the role of *effective rank*, *alignment* to Fisher eigendirections, and *projection fidelity*, respectively—yielding consistently lower drift than *LoRA*. GRIT further *matches or surpasses Orthogonal-LoRA, IA$^3$, DoRA/Eff-FT*, and *Shampoo* on the parameter-updates–versus–performance-retention frontier. Code repository.

## 1 PEFT's Blind Spot: Learning Inertia & Catastrophic Forgetting

Adapting billion–parameter LLMs stresses memory and bandwidth; PEFT addresses this by freezing most weights and training only a *small subset of additional parameters* (e.g., low–rank updates). Among PEFT variants, **LoRA** (Hu et al., 2021) and **QLoRA** (Dettmers et al., 2023) have become de facto standards. **The emerging trade–off**: Recent evidence that *"LoRA learns less and forgets less"* (Biderman et al., 2024) indicates that, relative to full fine–tuning, LoRA's low–rank update budget often yields *smaller gains on hard targets* (e.g., code/math) while *preserving* more of the base model's broad abilities (e.g., commonsense and general language competence). In short, PEFT often trades *peak task improvement* for *retention*: constrained adapters may underfit challenging distributions yet induce fewer high–impact shifts that erase pretraining knowledge.

### 1.1 Scaling Laws for Forgetting: LoRA

**Fine-tuning LLMs** invariably induces *catastrophic forgetting*—a drift away from the pretraining distribution that degrades general knowledge. In PEFT methods like LoRA, this forgetting is typ-

ically quantified by the increase in **pretraining loss** $L_{pt}$ after fine-tuning. Bethune et al. (2022) demonstrate that forgetting obeys a **power-law** with respect to the fine-tuning data volume $D_{\text{ft}}$ ( number of unique fine-tuning tokens) and the model size $N$ (number of model parameters):

$$L_{pt} = L_{pt}^0 + A\,\frac{D_{ft}^{\beta}}{N^{\alpha}} + E$$

where $L_{pt}^0$ is the original pretraining loss, and $A$, $\alpha$, $\beta$, $E$ are dataset- and model-specific constants. **This captures a key trade-off:** increasing $D_{ft}$ amplifies forgetting ($D_{ft}^{\beta}$), while larger models forget less due to $N^{-\alpha}$, *diluting per-update distortion across parameters.*

## 1.2 DIAGNOSIS: LEARNING INERTIA VS. HIGH-IMPACT UPDATES

**Forgetting is not just how much you train—it's *where* you move.** PEFT imposes *learning inertia* by freezing most weights and restricting adaptation to a low-rank subspace, yet interference still arises when updates overlap **high-curvature** modes of the pretraining objective. Let $L_{\text{pt}}(w)$ be the pretraining loss and let $\Delta w$ denote the PEFT-induced update. Near a well-fit solution, first-order terms are small and the **quadratic term** dominates:

$$\Delta L_{\text{pt}} \approx \tfrac{1}{2}\,\Delta w^\top H_{\text{pt}}\,\Delta w \;=\; \tfrac{1}{2}\sum_j \lambda_j \left(u_j^\top \Delta w\right)^2,$$

where $H_{\text{pt}} = \sum_j \lambda_j u_j u_j^\top$ is the **Hessian eigendecomposition**. **Intuition.** Forgetting is large when landscape is sharp (large $\lambda_j$) and when the update has large projections onto those sharp directions (large $|u_j^\top \Delta w|$) (Pascanu et al., 2013; Ghorbani et al., 2019; Keskar et al., 2017; Dinh et al., 2017).

**From principle to practice.** The quadratic form explains *why* forgetting increases but not *what* to monitor during training. We therefore introduce two *operational* geometry summaries that map directly onto the quadratic term and are simple to track online. **Two geometric amplifiers**:
*(i) Tail mass of updates.*

$$U_{\text{hi}}(\tau) \;=\; \sum_i \mathbf{1}\big(|\Delta w_i| > \tau\big),$$

the count of coordinates exceeding a magnitude threshold $\tau$. **Interpretation:** heavier tails imply more frequent large coordinates, increasing the chance that $|u_j^\top \Delta w|$ is large and thus amplifying the quadratic loss rise; this aligns with continual-learning evidence that large, concentrated steps drive interference (Kirkpatrick et al., 2017; Zenke et al., 2017; Aljundi et al., 2018; Chaudhry et al., 2019).

*(ii) Effective rank of the update covariance.* Let $\{\lambda_j^{(\Delta)}\}$ be the eigenvalues (descending) of $\mathbb{E}[\Delta w\,\Delta w^\top]$. Let's define:

$$r_{\text{eff}} \;=\; \min\Big\{k : \frac{\sum_{j=1}^k \lambda_j^{(\Delta)}}{\sum_j \lambda_j^{(\Delta)}} \geq \eta\Big\} \quad (\eta \in (0,1)).$$

**Interpretation:** larger $r_{\text{eff}}$ means update energy is spread across more directions, *raising the probability* of overlap with sharp Hessian modes and increasing $\sum_j \lambda_j(u_j^\top \Delta w)^2$ (Roy & Vetterli, 2007; Gavish & Donoho, 2014; Aghajanyan et al., 2021).

**Takeaway.** *Tail mass* controls *how big* the projections can be; *effective rank* controls *how many* directions those projections can land on. Either rising makes curvature overlap—and hence $\Delta L_{\text{pt}}$—more likely. This diagnosis motivates *geometry-aware* PEFT procedures that *shrink tails* and *concentrate rank* away from sharp modes.

**Evidence for the blind spot.** Large-scale evaluations show a **stable Pareto**: LoRA *learns less* than full fine-tuning on hard domains (code, math) yet *forgets less* of broad base abilities (HellaSwag, ARC-C, WinoGrande), while full FT induces **higher-rank** perturbations (often 10–100× typical LoRA ranks) (Biderman et al., 2024). This echoes **classic catastrophic interference** (McCloskey & Cohen, 1989; French, 1999) and **recent continual-learning views** for LLMs (Wu & Others, 2024; Vu & Others, 2022). Read through the scaling law lens in Sec. 1.1 – sets **how much** forgetting to

expect from $(D_{\text{ft}}, N)$, while the quadratic analysis explains **why** methods at the same budget diverge: update **geometry** multiplies the baseline via adapter-restricted curvature exposure $\overline{\kappa} = \text{tr}(PH_{\text{pt}}P)$ and by increasing functions of effective rank $\Phi(r_{\text{eff}})$ and tail mass $\Psi(U_{\text{hi}}(\tau))$. Standard, **geometry-agnostic** LoRA—first-order optimization in a fixed basis—tends to inflate $\overline{\kappa}$ and $U_{\text{hi}}(\tau)$ at fixed $(D_{\text{ft}}, N)$ (Hu et al., 2021; Biderman et al., 2024), motivating **curvature-aware PEFT** as the remedy.

**Implication.** If forgetting equals *(data/model)* times *(geometry/update)*, improving retention at fixed $(D_{\text{ft}}, N)$ requires **shrinking the geometry factor**. This motivates **geometry-aware PEFT**: estimate curvature in the adapter rank space and combine **natural-gradient/K-FAC** preconditioning with periodic **reprojection** toward high-signal, low-interference eigendirections (Amari, 1998; Martens & Grosse, 2015; Ollivier, 2015). In short, **geometry-aware PEFT is needed**, and we propose **GRIT**: *Geometry-Aware PEFT with K-FAC Preconditioning, Fisher-Guided Reprojection, and Dynamic Rank Adaptation*.

## 2 GRIT: GEOMETRY-AWARE PEFT WITH K-FAC PRECONDITIONING, FISHER-GUIDED REPROJECTION, AND DYNAMIC RANK ADAPTATION

*GRIT* is a geometry-aware PEFT framework that turns LoRA-style updates ($\Delta W = BA$) into a *dynamic, curvature-aligned* procedure through three coupled steps: **(i) curvature-aware preconditioning**—apply a K-FAC (Kronecker-Factored Approximate Curvature) Martens & Grosse (2015); Grosse & Martens (2016); Amari (1998) approximation to the Fisher $F$ *within the adapter subspace* to temper steps along sharp directions; **(ii) spectrum-driven rank scheduling**—read the per-layer Fisher eigenspectrum and *allocate rank where spectral energy concentrates*; **(iii) neural reprojection**—periodically *gate and align* the low-rank factors with the top-$k$ eigenspace of $F$ ($U_k U_k^\top$), preserving task progress while discarding drift. This loop repeats, so the adapter basis *tracks high-signal, low-interference directions* rather than remaining fixed. The formal objective balancing task loss with *curvature regularization* and *reprojection constraints* appears in **Fig. 1**, the end-to-end flow in **Fig. 2**, and the overall effect—*geometry-aligned*, **sparser**, and **more targeted** updates at the same memory budget—in **Fig. 3**.

$$\min_{A\in\mathbb{R}^{d\times r},\,B\in\mathbb{R}^{r\times d}} \underbrace{L_{\text{task}}(W_0 + BA)}_{\text{(1) Task Loss}} + \lambda_{\text{K}} \underbrace{\left\|F^{\frac{1}{2}}(BA)\right\|_F^2}_{\text{(2) Curvature Reg.}} + \lambda_{\text{R}} \underbrace{\left\|BA - U_k U_k^\top BA\right\|_F^2}_{\text{(3) Reprojection Reg.}}$$

Figure 1: **GRIT Objective: Curvature-Aware, Projection-Constrained Fine-Tuning.** This loss balances task performance with geometric awareness and subspace filtering: **(1) Task loss** $L_{\text{task}}(W_0 + BA)$ optimizes the instruction-tuning objective using the low-rank update $BA$; **(2) Curvature regularization** $\|F^{\frac{1}{2}}BA\|_F^2$ penalizes updates in high-sensitivity regions defined by the Fisher matrix $F$, promoting safe adaptation; **(3) Reprojection regularization** $\|BA - U_k U_k^\top BA\|_F^2$ encourages the update to remain within the subspace spanned by the top-$k$ eigenvectors of $F$, filtering noisy or low-impact directions. Hyperparameters $\lambda_{\text{K}}$ and $\lambda_{\text{R}}$ control the curvature and reprojection terms, enabling geometry-respecting fine-tuning.

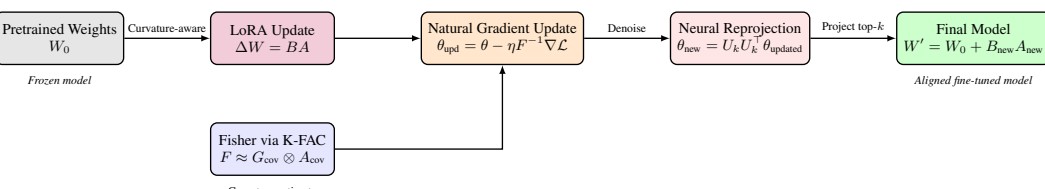

Figure 2: **GRIT Geometry-Aware Fine-Tuning Pipeline.** Starting from frozen pretrained weights $W_0$, GRIT applies a low-rank update $\Delta W = BA$ using LoRA. The Fisher Information Matrix $F$ is approximated using K-FAC to compute a natural gradient update in curvature-sensitive directions. This is followed by a projection onto the dominant eigen-subspace of $F$ via $\theta_{\text{new}} = U_k U_k^\top \theta_{\text{updated}}$, producing the refined update $\Delta W_{\text{new}} = B_{\text{new}} A_{\text{new}}$. The final model becomes $W' = W_0 + \Delta W_{\text{new}}$, incorporating only aligned, geometry-aware directions.

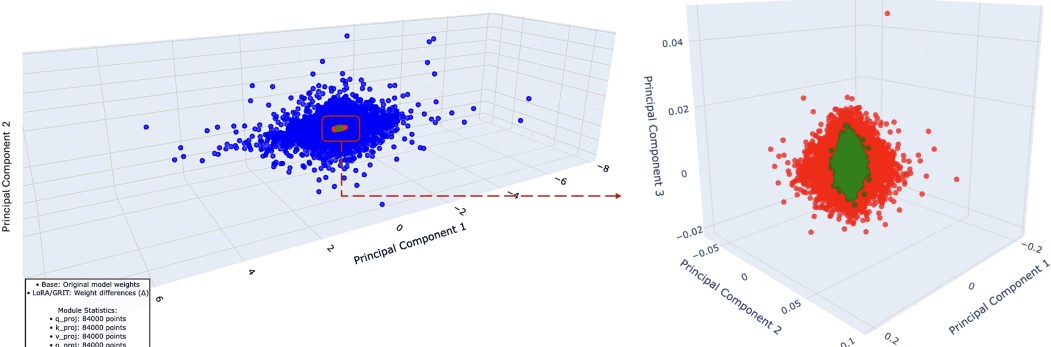

**(a)** Global view: original model weights vs. LoRA/GRIT $\Delta$-weights in PCA space.

**(b)** Zoom near the origin comparing LoRA vs. GRIT $\Delta$-weights only.

Figure 3: **Geometry of parameter updates: GRIT concentrates $\Delta$-weights into curvature-aligned subspaces.** *Setup.* PCA on parameter vectors from attention projections $(q_{\text{proj}}, k_{\text{proj}}, v_{\text{proj}}, o_{\text{proj}})$ across layers; points are mean-centered and embedded into the leading PCs (no extra scaling). **(a) Overall (left).** *Blue* points depict the **entire model's parameter space** as realized by the *original (base) weights*. Superimposed at the center, *red* points are **LoRA** *update deltas* ($\Delta W$), and *green* points are **GRIT** *update deltas*. The visual shows that the full base space is broad and anisotropic, while both LoRA and GRIT operate in a much *smaller central region*—the effective fine-tuning manifold. **(b) Zoom on $\Delta$ (right).** The base cloud is omitted to compare updates directly: *red* = LoRA $\Delta W$, *green* = GRIT $\Delta W$. GRIT forms a *tighter, ellipsoidal core* with reduced radial spread versus LoRA, consistent with *rank-space natural-gradient* preconditioning and *Fisher-aligned reprojection* that bias updates toward high-curvature eigendirections while suppressing diffuse, low-signal axes. **Interpretation.** GRIT densifies signal in a low-rank, curvature-aware subspace (smaller support, tighter covariance) without enlarging the update footprint—steering limited parameters toward directions that matter for stability and generalization.

## 2.1 LOW-RANK ADAPTATION SETUP

Consider a transformer module with a linear projection parameterized by a weight matrix $W \in \mathbb{R}^{d_{\text{out}} \times d_{\text{in}}}$. *Full fine-tuning* updates all entries of $W$, which is prohibitive at scale. *Low-rank adaptation* (LoRA/QLoRA) instead introduces a parameter-efficient update

$$\Delta W = BA, \qquad B \in \mathbb{R}^{d_{\text{out}} \times r}, \ A \in \mathbb{R}^{r \times d_{\text{in}}}, \ r \ll \min(d_{\text{in}}, d_{\text{out}}),$$

so the effective weight is $W' = W + \alpha \Delta W$ (with a small scaling $\alpha$). This reduces trainable parameters from $O(d_{\text{out}} d_{\text{in}})$ to $O(r(d_{\text{in}} + d_{\text{out}}))$, preserving *expressivity per parameter* while keeping memory/compute manageable.

***Caveat—geometry agnosticism.*** Standard LoRA learns $A, B$ in a **fixed** low-rank subspace using first-order updates, *ignoring* the local curvature of the loss. As a result, steps can over-expose sharp directions of the pretraining objective, amplifying interference. **GRIT** eliminates this blind spot by making the low-rank subspace itself *geometry-aware* via three coupled components: **curvature-aware preconditioning**, **Fisher-guided reprojection**, and **dynamic rank adaptation**.

## 2.2 K-FAC–BASED PRECONDITIONING

**From gradients to *natural* gradients.** Raw stochastic gradients need not align with the loss geometry. The *natural gradient* rescales $\nabla_\theta \mathcal{L}$ by the inverse Fisher information matrix (FIM), following steepest descent under the KL metric (Amari, 1998):

$$\theta_{t+1} = \theta_t - \eta F^{-1} \nabla_\theta \mathcal{L}(\theta_t), \quad F = \mathbb{E}\big[\nabla \log p(x; \theta) \nabla \log p(x; \theta)^\top\big].$$

Directly forming/inverting $F$ is infeasible for LLMs (quadratic storage, cubic inversion).

**K-FAC for layers, *restricted* to rank space.** K-FAC approximates the layerwise Fisher for a linear map $y = Wx$ by a Kronecker product of second moments of the *input activations* and *output gradients* (Martens & Grosse, 2015; Grosse & Martens, 2016):

$$F_{\text{layer}} \approx \Sigma_g \otimes \Sigma_a, \quad \Sigma_a = \mathbb{E}[x\,x^\top], \ \Sigma_g = \mathbb{E}[g\,g^\top], \ g \equiv \tfrac{\partial \mathcal{L}}{\partial y}.$$

Then the *natural-gradient* preconditioned weight update admits the efficient form

$$\nabla W_{\text{nat}} \approx \Sigma_g^{-1} \nabla W \Sigma_a^{-1},$$

avoiding an explicit inversion of $F_{\text{layer}}$.

**Rank-space K-FAC for LoRA.** For the low-rank update $\Delta W = BA$, GRIT applies K-FAC *within* the rank-$r$ subspace spanned by $A, B$. Let $x \in \mathbb{R}^{d_{\text{in}}}$ denote input activations and $g \in \mathbb{R}^{d_{\text{out}}}$ the backpropagated gradients at the pre-activation. We define the rank-projected statistics

$$a_r = A x \in \mathbb{R}^r, \qquad g_r = B^\top g \in \mathbb{R}^r,$$

and maintain the *rank-space* covariances

$$\Sigma_a^{(r)} = \mathbb{E}[a_r a_r^\top] \in \mathbb{R}^{r \times r}, \qquad \Sigma_g^{(r)} = \mathbb{E}[g_r g_r^\top] \in \mathbb{R}^{r \times r}.$$

Under the standard K-FAC independence approximation, the Fisher *restricted to the LoRA subspace* factorizes as

$$F_{\text{rank}} \approx \Sigma_g^{(r)} \otimes \Sigma_a^{(r)}.$$

Consequently, the preconditioned gradient for the low-rank update satisfies

$$\nabla(\Delta W)_{\text{nat}} \approx \Sigma_g^{(r)\,-1} \nabla(\Delta W) \Sigma_a^{(r)\,-1},$$

which *decouples* into factor-wise updates (implementation-conformant for $\Delta W = BA$):

$$\boxed{\nabla B \leftarrow \nabla B \Sigma_g^{(r)\,-1}, \qquad \nabla A \leftarrow \Sigma_a^{(r)\,-1} \nabla A.}$$

Intuitively, $\Sigma_g^{(r)\,-1}$ suppresses steps along *high-curvature output directions* (sharp modes), while $\Sigma_a^{(r)\,-1}$ *removes input-scale anisotropy* in the adapter subspace—yielding **curvature-aligned**, *scale-invariant* updates.

**Practicalities.** To ensure numerical stability, GRIT uses (i) *damped* covariances $\tilde{\Sigma}_a^{(r)} = \Sigma_a^{(r)} + \lambda_a I$, $\tilde{\Sigma}_g^{(r)} = \Sigma_g^{(r)} + \lambda_g I$; (ii) *streaming* (EMA) estimates with burn-in before inversion; and (iii) efficient solves via Cholesky on $r \times r$ matrices (with $r \ll d_{\text{in}}, d_{\text{out}}$). All statistics are computed per layer and can be offloaded/cached across devices. In GRIT, K-FAC preconditioning is the **first** step: subsequent *Fisher-guided reprojection* rotates $(A, B)$ toward dominant eigendirections and *dynamic rank scheduling* allocates capacity where the Fisher spectrum has mass (see **Fig. 1** and **Fig. 2**).

**Takeaway.** K-FAC in rank space gives GRIT a *computationally light* approximation to the natural gradient—retaining key **second-order** benefits (curvature awareness, KL geometry) while scaling to billion-parameter LLMs (Amari, 1998; Martens & Grosse, 2015; Grosse & Martens, 2016).

## 2.3 NEURAL REPROJECTION

**Motivation.** Preconditioning corrects *step directions* but leaves the *update subspace* fixed. Over training, the LoRA subspace spanned by $A$ and $B$ (rank $r$) can drift, accumulate redundancy, or misalign with **high-curvature** directions—wasting gradient signal and inflating interference. *Neural reprojection* remedies this by **reshaping** the subspace itself to track informative curvature.

**Curvature-aligned subspace.** Let the rank-space covariances (Sec. K-FAC) be

$$\Sigma_a^{(r)} = \mathbb{E}[a_r a_r^\top], \qquad \Sigma_g^{(r)} = \mathbb{E}[g_r g_r^\top] \in \mathbb{R}^{r \times r},$$

with eigendecompositions

$$\Sigma_a^{(r)} = U_A \Lambda_A U_A^\top, \qquad \Sigma_g^{(r)} = U_G \Lambda_G U_G^\top,$$

where $U_A, U_G \in \mathbb{R}^{r \times r}$ are orthogonal and $\Lambda_A, \Lambda_G \succeq 0$ carry *curvature energy* per direction. Let $U_A^{(k)}$ and $U_G^{(k)}$ collect the top-$k$ eigenvectors (by eigenvalue), and define projectors

$$P_A = U_A^{(k)}(U_A^{(k)})^\top, \qquad P_G = U_G^{(k)}(U_G^{(k)})^\top \in \mathbb{R}^{r \times r}.$$

We **reproject** the LoRA factors onto these dominant eigenspaces:

$$\boxed{A \leftarrow P_A A, \qquad B \leftarrow B P_G}$$

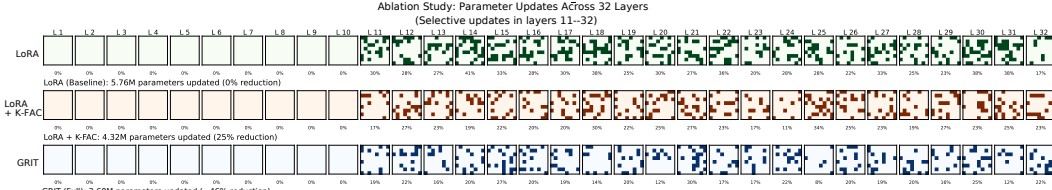

Figure 4: **Ablation—*parameter update patterns*** across LLaMA (32 layers). Each **8×8** mini-grid depicts one layer; *colored cells* mark updated parameters. We freeze **layers 1–10** because these early layers encode *general, task-agnostic representations*, while **layers 10–15** serve as *transition* to task-specific features and **layers 16–32** provide *refinement* (Zhao et al., 2024); this follows the PEFT practice of adapting only the **middle/later layers** (**11–32**). Within adapted layers, the mean update density is **LoRA** *30%* (green), **LoRA+K-FAC** *22.5%* (orange), and **GRIT** *18.75%* (blue). *Note:* these are **per-layer** densities; the **overall** update fraction is smaller because early layers are *not* adapted. Totals (annotated under each row): **LoRA** *5.76M* params (0% reduction), **LoRA+K-FAC** *4.32M* (25% reduction), **GRIT** *3.60M* (37.5% reduction). Counts assume each adapted layer updates **4** weight matrices of **65,536** parameters. *Takeaway:* **curvature-aware** preconditioning and **neural reprojection** *reduce update density* vs. standard LoRA while retaining adaptation capacity.

so the low-rank update $\Delta W = BA$ remains in a *rank-$r$* space with a **curvature-aligned** basis.

**Gating, scheduling, and stability.** To avoid premature rotations, we enable the $G$-side projection only after $\Sigma_g^{(r)}$ has accumulated at least $N_{\min}$ effective samples; otherwise, we *fallback* to $U_A^{(k)}$ for both sides in the first epochs. Reprojection runs at a fixed frequency (every $T$ steps) or adaptively when the spectrum mass ratio $\sum_{i=1}^{k} \Lambda_{(\cdot),i} / \sum_{i=1}^{r} \Lambda_{(\cdot),i}$ crosses a threshold. Since $\Sigma_a^{(r)}$ and $\Sigma_g^{(r)}$ are $r \times r$, eigendecompositions are *cheap*, and $U_A^{(k)}, U_G^{(k)}$ are cached between steps. For numerical robustness, we use $\tilde{\Sigma}_{(\cdot)}^{(r)} = \Sigma_{(\cdot)}^{(r)} + \lambda I$ (damping), warm up statistics before projection, and interpolate updates if needed:

$$A \leftarrow (1-\gamma)A + \gamma P_A A, \qquad B \leftarrow (1-\gamma)B + \gamma B P_G, \quad \gamma \in [0,1].$$

**Effect.** Neural reprojection *removes low-energy directions*, suppresses noise, and **rotates** the LoRA subspace toward *high-signal, low-interference* eigendirections. Unlike pure preconditioning (which only rescales steps), reprojection **evolves the basis** so that adaptation occurs where curvature indicates capacity is most valuable—improving **stability** and **parameter efficiency** at fixed rank.

## 2.4 DYNAMIC RANK ADAPTATION

**Why adapt rank?** Neural reprojection aligns the subspace, but a *fixed* rank $r$ can still **underfit** (too few directions) or **overfit** (retain redundant, low-energy directions). GRIT therefore makes the *effective* rank evolve with the spectrum.

**Energy-based rule.** Let $\lambda_1 \geq \cdots \geq \lambda_r \geq 0$ be eigenvalues of the rank-space covariance (activation- or Fisher-side; cf. Secs. 1.2, K-FAC). Define the smallest integer $k$ capturing a target energy fraction $\tau \in (0,1]$:

$$k = \min \left\{ j \ \middle| \ \frac{\sum_{i=1}^{j} \lambda_i}{\sum_{i=1}^{r} \lambda_i} \geq \tau \right\}.$$

Here, "*energy*" is cumulative variance/mass in the leading eigendirections; larger $\lambda_i$ encode *high-signal* axes, smaller ones *noise* or redundancy.

**Constraints and gating.** We bound the effective rank by

$$k \in [\texttt{min\_rank}, r],$$

where $\texttt{min\_rank}$ prevents collapse and $r$ is the max LoRA rank at initialization. To avoid early misestimation, we apply a *warmup gate*: adaptive updates to $k$ start only after sufficient samples accumulate for stable spectra (e.g., Fisher/EMA burn-in).

**How adaptation is realized.** Rank adaptation is enacted through the projection operators used in *neural reprojection*: with $U^{(k)}$ the top-$k$ eigenvectors, the projectors $P_A = U_A^{(k)}(U_A^{(k)})^\top$, $P_G =$

$U_G^{(k)}(U_G^{(k)})^\top$ remove low-energy directions during

$$A \leftarrow P_A A, \qquad B \leftarrow B P_G.$$

No hard masking or tensor resizing is required—the suppressed directions remain *stored* and can re-enter if their eigenvalues grow later. This yields a **lightweight, reversible** mechanism with minimal overhead.

**Update incorporation.** After reprojection and rank selection, the refined low-rank update is

$$\Delta W_{\text{new}} \;=\; B_{\text{new}} A_{\text{new}}, \qquad W' \;=\; W \;+\; \Delta W_{\text{new}},$$

providing *curvature-aware*, rank-adaptive fine-tuning while preserving a stable parameterization of the adapters.

**Implementation summary.** Algorithm 1 maps directly to our system: rank-space covariances via autograd hooks; sample-gated, damped K-FAC inverses; natural-gradient preconditioning before the optimizer step; and periodic Fisher-guided reprojection with dynamic rank. See Appendix C for background and Appendix D for derivations.

---

**Algorithm 1** GRIT training loop

---

**Require:** Pretrained weights $W_0$; LoRA rank $r$; LoRA scaling $\alpha$; data stream $\mathcal{D}$; learning rate $\eta$; damping $\lambda$; frequencies (`kfac_upd`, `reproj_freq`); thresholds (`kfac_min`, $\tau$, `min_rank`); flags (`ng_warmup`, `reproj_warmup`, `use_two_sided`, `rank_adapt`)

1: Initialize LoRA factors $\{A_\ell, B_\ell\}$; set $A_{\text{cov}} \leftarrow I, G_{\text{cov}} \leftarrow I$; *inv_ready* $\leftarrow$ **False**; $n_{\text{cov}} \leftarrow 0$; *step* $\leftarrow 0$
2: **for** each minibatch $(x, y) \in \mathcal{D}$ **do**
3:      Forward with $W' = W_0 + \alpha \sum_\ell B_\ell A_\ell$; compute loss $\mathcal{L}$
4:      Backprop to obtain raw gradients $\nabla A_\ell, \nabla B_\ell$ and per-layer tensors $X, \delta Y$
5:      **if** *step* $\geq$ `ng_warmup` **then**
6:          **if** *step* mod `kfac_upd` $= 0$ **then**
7:              $a_r \leftarrow X A_\ell^\top$; $g_r \leftarrow \delta Y B_\ell$                                  ▷ rank-space stats
8:              Accumulate: $A_{\text{cov}} \leftarrow A_{\text{cov}} + a_r a_r^\top$; $G_{\text{cov}} \leftarrow G_{\text{cov}} + g_r g_r^\top$; $n_{\text{cov}} \leftarrow n_{\text{cov}} + 1$
9:              **if** $n_{\text{cov}} \geq$ `kfac_min` **then**
10:                 Compute $(A_{\text{cov}} + \lambda I)^{-1}$ and $(G_{\text{cov}} + \lambda I)^{-1}$; *inv_ready* $\leftarrow$ **True**
11:              **end if**
12:          **end if**
13:          **if** *inv_ready* **then**
14:              Natural gradient: $\nabla B_\ell \leftarrow \nabla B_\ell G_{\text{cov}}^{-1}$; $\nabla A_\ell \leftarrow A_{\text{cov}}^{-1} \nabla A_\ell$
15:          **end if**
16:      **end if**
17:      Optimizer step on $\{A_\ell, B_\ell\}$; freeze $W_0$
18:      **if** *step* $\geq$ `reproj_warmup` **and** *step* mod `reproj_freq` $= 0$ **then**
19:          Eigendecompose $A_{\text{cov}} = U_A \Lambda_A U_A^\top$
20:          **if** `use_two_sided` **and** *inv_ready* **then**
21:              Eigendecompose $G_{\text{cov}} = U_G \Lambda_G U_G^\top$
22:          **end if**
23:          **if** `rank_adapt` **then**
24:              $k \leftarrow \min\left\{j \mid \sum_{i=1}^{j} \lambda_i / \sum_{i=1}^{r} \lambda_i \geq \tau\right\}$; $k \leftarrow \max(k, \text{min\_rank})$
25:          **else**
26:              $k \leftarrow$ `reproj_k`
27:          **end if**
28:          $A_\ell \leftarrow U_A^{(k)} U_A^{(k)\top} A_\ell$
29:          $B_\ell \leftarrow \begin{cases} B_\ell U_G^{(k)} U_G^{(k)\top}, & \text{if } \texttt{use\_two\_sided} \text{ and } inv\_ready \\ B_\ell U_A^{(k)} U_A^{(k)\top}, & \text{otherwise} \end{cases}$
30:      **end if**
31:      *step* $\leftarrow$ *step* + 1
32: **end for**
33: **return** $W' = W_0 + \alpha \sum_\ell B_\ell A_\ell$

---

## 3 EXPERIMENTS AND RESULTS

**Setup & Datasets.** Unless stated, we fine-tune **Llama 3.2–3B** and **Llama 3.1–8B** with *4-bit NF4* quantization and `bf16` compute. **GRIT** reuses the **QLoRA** data pipeline for an apples-to-apples comparison; *reprojection* is gated by curvature-sample thresholds, and *dynamic rank* follows a cumulative-energy rule. Seeds, optimizers, and schedules appear in Appx. B.8. Evaluation spans five benchmarks: *Alpaca* (Wang et al., 2023) (52k instruction–response pairs), *Dolly 15k* (Databricks, 2023) (15k human-written prompts), *BoolQ* (Clark et al., 2019) (yes/no over Wikipedia), *QNLI* from GLUE (Wang et al., 2019) (sentence-pair entailment), and *GSM8K* (Cobbe et al., 2021) (grade-school math reasoning).

Table 1: **Extended baselines on *Llama-3.2 3B***. Rows are grouped by dataset (blue/gray bands). Each metric cell reports the *absolute* score followed by a *relative delta*: for all columns except **GRIT**, the delta is computed *vs. GRIT* (↑= higher than GRIT, ↓= lower than GRIT); in the **GRIT** column, the delta is computed *vs. LoRA* to show GRIT's improvement or drop relative to the canonical baseline. For ROUGE/BLEU/BERTScore/Accuracy/Precision/Recall/F1, larger is better; arrows therefore indicate better/worse accordingly. **Bold** marks the best value within the row. The "# Params Trained" rows report the *absolute* number of trainable parameters for each method and, in parentheses, the % change *vs. LoRA* (lower is better). This layout makes quality deltas relative to GRIT explicit while simultaneously exposing GRIT's parameter savings over LoRA.

| Model: Llama-3.2 (3B) | LoRA | QLoRA | GRIT (vs. LoRA) | Orthogonal-LoRA | IA$^3$ | DoRA/Eff-FT | Shampoo |
|---|---|---|---|---|---|---|---|
| **ALPACA** | | | | | | | |
| ROUGE-1 | 0.1852 | 0.1292 | **0.1844** (↓ 0.8%) | 0.1870 (↑ 1.4%) | 0.1680 (↓ 8.9%) | **0.1915** (↑ 3.9%) | 0.1885 (↑ 2.2%) |
| ROUGE-2 | 0.0825 | 0.0562 | **0.0818** (↓ 0.8%) | 0.0836 (↑ 2.2%) | 0.0710 (↓ 13.2%) | **0.0868** (↑ 6.1%) | 0.0850 (↑ 3.9%) |
| ROUGE-L | 0.1426 | 0.0983 | **0.1425** (↓ 0.1%) | 0.1440 (↑ 1.1%) | 0.1310 (↓ 8.1%) | **0.1478** (↑ 3.7%) | 0.1456 (↑ 2.2%) |
| BLEU | 0.0443 | 0.0235 | **0.0430** (↓ 2.9%) | 0.0451 (↑ 4.9%) | 0.0380 (↓ 11.6%) | **0.0472** (↑ 9.8%) | 0.0461 (↑ 7.2%) |
| BERTScore | 0.8343 | 0.7948 | **0.8354** (↑ 0.1%) | 0.8350 (↓ 0.0%) | 0.8230 (↓ 1.5%) | 0.8349 (↓ 0.1%) | 0.8351 (↓ 0.0%) |
| # Params Trained | 24.31M | 24.31M | **8.45M** (65.3% ↓) | 24.31M (0.0%) | **2.10M** (91.4% ↓) | 24.31M (0.0%) | 24.31M (0.0%) |
| **Dolly-15k** | | | | | | | |
| ROUGE-1 | 0.1733 | 0.1108 | **0.1976** (↑ 14.0%) | 0.1795 (↓ 9.2%) | 0.1602 (↓ 18.9%) | 0.1931 (↓ 2.3%) | 0.1864 (↓ 5.7%) |
| ROUGE-2 | 0.0824 | 0.0519 | **0.0994** (↑ 20.7%) | 0.0856 (↓ 13.9%) | 0.0718 (↓ 27.8%) | **0.1006** (↑ 1.2%) | 0.0941 (↓ 5.3%) |
| ROUGE-L | 0.1368 | 0.0884 | **0.1568** (↑ 14.6%) | 0.1402 (↓ 10.6%) | 0.1243 (↓ 20.7%) | 0.1541 (↓ 1.7%) | 0.1486 (↓ 5.2%) |
| BLEU | 0.0533 | 0.0297 | **0.0560** (↑ 5.1%) | 0.0542 (↓ 3.2%) | 0.0451 (↓ 19.5%) | **0.0574** (↑ 2.5%) | 0.0566 (↑ 1.1%) |
| BERTScore | 0.8295 | 0.8005 | **0.8344** (↑ 0.6%) | 0.8311 (↓ 0.4%) | 0.8192 (↓ 1.8%) | 0.8335 (↓ 0.1%) | 0.8322 (↓ 0.3%) |
| # Params Trained | 24.31M | 24.31M | **17.01M** (30.0% ↓) | 24.31M (0.0%) | **2.10M** (91.4% ↓) | 24.31M (0.0%) | 24.31M (0.0%) |
| **GSM8K** | | | | | | | |
| ROUGE-1 | 0.5582 | 0.5518 | 0.5532 (↓ 0.9%) | 0.5594 (↑ 1.1%) | 0.5401 (↓ 2.4%) | **0.5610** (↑ 1.4%) | 0.5601 (↑ 1.2%) |
| ROUGE-2 | 0.3236 | 0.3197 | 0.3173 (↓ 1.9%) | 0.3249 (↑ 2.4%) | 0.3050 (↓ 3.9%) | **0.3268** (↑ 3.0%) | 0.3257 (↑ 2.6%) |
| ROUGE-L | 0.5228 | 0.5169 | 0.5167 (↓ 1.2%) | 0.5233 (↑ 1.3%) | 0.5078 (↓ 1.7%) | **0.5241** (↑ 1.4%) | 0.5237 (↑ 1.4%) |
| Accuracy | 0.3935 | 0.3836 | 0.3867 (↓ 1.7%) | 0.3962 (↑ 2.5%) | 0.3564 (↓ 7.8%) | **0.3991** (↑ 3.2%) | 0.3978 (↑ 2.9%) |
| # Params Trained | 24.31M | 24.31M | **15.30M** (37.1% ↓) | 24.31M (0.0%) | **2.10M** (91.4% ↓) | 24.31M (0.0%) | 24.31M (0.0%) |
| **QNLI** | | | | | | | |
| Accuracy | 0.8938 | 0.8885 | **0.9053** (↑ 1.3%) | 0.8984 (↓ 0.8%) | 0.8820 (↓ 2.6%) | 0.9026 (↓ 0.3%) | 0.9001 (↓ 0.6%) |
| Precision | 0.8939 | 0.8971 | **0.9059** (↑ 1.3%) | 0.8991 (↓ 0.8%) | 0.8842 (↓ 2.4%) | 0.9037 (↓ 0.2%) | 0.9010 (↓ 0.5%) |
| Recall | 0.8939 | 0.8893 | **0.9055** (↑ 1.3%) | 0.8977 (↓ 0.9%) | 0.8805 (↓ 2.8%) | 0.9021 (↓ 0.4%) | 0.8990 (↓ 0.7%) |
| F1 | 0.8938 | 0.8880 | **0.9052** (↑ 1.3%) | 0.8981 (↓ 0.8%) | 0.8810 (↓ 2.7%) | 0.9029 (↓ 0.3%) | 0.8996 (↓ 0.6%) |
| # Params Trained | 24.31M | 24.31M | **7.75M** (68.1% ↓) | 24.31M (0.0%) | **1.20M** (95.1% ↓) | 24.31M (0.0%) | 24.31M (0.0%) |
| **BoolQ** | | | | | | | |
| Accuracy | 0.7834 | 0.7525 | **0.7749** (↓ 1.1%) | 0.7860 (↑ 1.4%) | 0.7402 (↓ 4.5%) | 0.7815 (↑ 0.9%) | 0.7851 (↑ 1.3%) |
| Precision | 0.7982 | 0.7491 | **0.7908** (↓ 0.9%) | 0.8010 (↑ 1.3%) | 0.7385 (↓ 6.6%) | 0.7972 (↑ 0.8%) | 0.7994 (↑ 1.1%) |
| Recall | 0.8720 | 0.9050 | **0.8671** (↓ 0.6%) | 0.8692 (↑ 0.2%) | 0.8204 (↓ 5.4%) | 0.8701 (↑ 0.3%) | 0.8710 (↑ 0.4%) |
| F1 | 0.8335 | 0.8197 | **0.8272** (↓ 0.8%) | 0.8357 (↑ 1.0%) | 0.7796 (↓ 5.8%) | 0.8324 (↑ 0.6%) | 0.8348 (↑ 0.9%) |
| # Params Trained | 24.31M | 24.31M | **15.03M** (38.2% ↓) | 24.31M (0.0%) | **1.60M** (93.4% ↓) | 24.31M (0.0%) | 24.31M (0.0%) |

**Baselines.** We compare GRIT to strong PEFT baselines: **LoRA** (Hu et al., 2021), **QLoRA** (Dettmers et al., 2023), **IA**$^3$ (Liu et al., 2022) (per-module gating in lieu of rank updates), **DoRA** (Liu et al., 2024) (direction–magnitude decomposition for stabler adapters), and an **orthogonal-LoRA** control (basis orthogonality). To isolate the role of curvature modeling apart from subspace alignment, we include factored second-order **Shampoo** (Anil et al., 2021) *without* Fisher-guided reprojection as an optimizer control. This set spans where capacity is injected (ranks vs. gates), how it is constrained (quantization, decomposition, orthogonality), and how updates are preconditioned (first- vs. second-order), providing a clean backdrop for GRIT's geometry-aware contributions.

**Performance.** **GRIT matches or exceeds quality while sharply cutting trainable parameters.** Across five benchmarks and two model sizes, GRIT attains parity or small gains over strong PEFT baselines while *substantially* reducing update footprint (**Table 1**). On **Alpaca**, GRIT is within <1% of the best ROUGE-1/2/L and BERTScore at both scales, yet trains only **8.45M** params on 3B (**65.3%** ↓ vs. LoRA/QLoRA) and **19.27M** on 8B (**77.0%** ↓). On **Dolly-15k**, GRIT attains the top ROUGE-1/2/L and BLEU for 3B, with a **30.0%** reduction in trained params (17.01M vs. 24.31M), and remains competitive on 8B while trimming **35.2%–54.5%**. On **GSM8K** (reasoning), GRIT ties or trails by < 1.5% on sequence metrics for 3B, and *wins accuracy* on 8B (**0.6619**, best-in-block) with a **27.8%** reduction (60.5M). On **QNLI** (NLI), GRIT is best or within < 1.0% across Accuracy/Precision/Recall/F1 at 3B while cutting params by **68.1%** (7.75M), and remains near the block leader on 8B with **64.9%–80.0%** savings. On **BoolQ**, GRIT is at or near the top across metrics with **38.2%** (3B) and **26.6%** (8B) fewer trained parameters. An extended ablation on Llama-3.2 3B (Table 1) confirms the trend against stronger PEFT/optimizer controls: orthogonal-LoRA, IA$^3$,

DoRA/Eff-FT, and Shampoo show modest metric fluctuations around GRIT, but none close the parameter-efficiency gap.

**Where the savings come from.** GRIT's dynamic, geometry-aware allocation adapts the *effective* rank and concentrates updates in informative layers, yielding task-dependent compression rather than a fixed adapter budget. This appears as consistent per-task reductions in the "# Params Trained" column of **Table 1** (e.g., **65.3**% ↓–**77.0**% ↓ on Alpaca; **30.0**% ↓–**54.5**% ↓ on Dolly-15k; **26.6**% ↓–**80.0**% ↓ on BoolQ/QNLI) while maintaining block-best or near-best quality. The update-pattern visualization in **Fig. 4** shows the same story at the layer level: GRIT suppresses early layers and densifies updates selectively in middle-to-late blocks, delivering *sparser, better-aimed* updates at the same memory budget. Overall, **GRIT offers a favorable quality–efficiency trade-off** relative to LoRA/QLoRA and stronger PEFT baselines, with *consistent parameter savings and competitive accuracy* across instruction following, classification, and math reasoning.

### 3.1 SCALING LAWS FOR FORGETTING: LoRA VS. GRIT

**Baseline intuition.** Forgetting after fine-tuning follows a power law in the amount of new data and the model size (Bethune et al., 2022). See Section 1.1.

**Scaling law of forgetting for GRIT.** Standard LoRA optimizes in a fixed low-rank basis, which can accidentally point updates into high-curvature directions that amplify drift. GRIT adds *curvature-aware preconditioning* and *Fisher-guided reprojection*, which (i) shrink steps along sharp modes and (ii) rotate the low-rank basis toward informative eigendirections. The net effect is an *effective capacity multiplier* $\Xi_{\mathrm{GRIT}} > 1$ in the denominator:

$$L_{pt}^{\mathrm{GRIT}} = L_{pt}^0 + A \frac{D_{ft}^{\beta}}{\left(\Xi_{\mathrm{GRIT}} N\right)^{\alpha}} + E, \quad \Xi_{\mathrm{GRIT}} = (1 + \gamma_r r_{\mathrm{eff}})(1 + \gamma_a \rho_{\mathrm{align}})(1 + \gamma_p \pi_{\mathrm{proj}})$$

We parameterize $\Xi_{\mathrm{GRIT}}$ by measurable geometry where $r_{\mathrm{eff}}$ is the adapter's *effective rank* (usable capacity), $\rho_{\mathrm{align}} \in [0,1]$ is alignment with the Fisher top-$k$ subspace (curvature alignment), and $\pi_{\mathrm{proj}} \in [0,1]$ is the spectral mass retained after reprojection (signal concentration). The scalings $\gamma_{\{\cdot\}} \geq 0$ are fitted from runs that vary $D_{ft}$, rank, and reprojection frequency. See the full derivation in Appendix Section E.4.

**How to read the law.** At fixed $(D_{ft}, N)$, improving any of $\{r_{\mathrm{eff}}, \rho_{\mathrm{align}}, \pi_{\mathrm{proj}}\}$ increases $\Xi_{\mathrm{GRIT}}$ and lowers $L_{pt}^{\mathrm{GRIT}}$. Practically, we recommend reporting *Fisher spectra*, *effective ranks*, and *alignment proxies* alongside task quality so the geometry term is auditable at scale. For background on natural-gradient/K-FAC curvature handling underlying the intuition, see Amari (1998); Martens & Grosse (2015); the learn-vs-forget trade-offs specific to PEFT are discussed in Bethune et al. (2022); Biderman et al. (2024).

## 4 CONCLUSION

**What we did.** We introduced **GRIT**, a *geometry-aware* PEFT recipe that augments LoRA with three synergistic components: *rank-space natural gradients* via K-FAC, *Fisher-guided reprojection* to align updates with dominant curvature, and *dynamic rank adaptation* to allocate capacity where signal concentrates. Together, these mechanisms steer low-rank updates into *high-signal, low-interference* directions.

**What we achieved.** Across instruction-following, classification, and reasoning tasks on LLaMA backbones, GRIT *matches or exceeds* strong baselines while *substantially reducing* trainable parameters (typ. ~46% average, 25–80% across tasks), yielding a robust efficiency–quality trade-off. Empirically, GRIT's curvature-modulated forgetting obeys a power-law with a larger effective capacity factor, indicating consistently lower drift at fixed data and model size.

**What's next.** Future work includes tighter curvature estimators beyond rank-space K-FAC, principled schedules for reprojection frequency and rank budgets, broader evaluations on multimodal and retrieval-augmented backbones, and end-to-end tooling for geometry telemetry—toward scalable, reliable PEFT under real-world constraints.

## REPRODUCIBILITY STATEMENT

**Scope and artifacts.** We release *all* artifacts required to exactly reproduce our results: source code, training/evaluation scripts, configuration files (`.yaml`), environment files (`environment.yml` and `requirements.txt`), experiment manifests (`.jsonl`), random seeds, and raw evaluation outputs. The repository contains a `Makefile` with recipes for data preparation, training, checkpointing, and evaluation. We also include a `REPORT.md` that records command lines, wall-clock times, GPU memory, and commit hashes for every run.

**Hardware.** All experiments were run on NVIDIA A100 80GB GPUs (SXM4), except *Llama 3.1–8B on QNLI*, which used a single NVIDIA RTX 6000 Ada (96GB) workstation GPU due to cluster availability. Each run used mixed precision on tensor cores. Host CPUs were dual Intel Xeon Silver 4314 or AMD EPYC 7452; system RAM $\geq$ 256 GB. Experiments were orchestrated with `slurm` and `torchrun`.

**Software environment.** We provide an exact, pinned software stack and export a `conda` environment file. Reproducing on different CUDA/cuDNN versions is typically benign but may cause $\pm$0.1–0.3pt jitter in text metrics due to kernel and RNG differences.

Table 2: **Environment & Framework Versions (pinned)**

| Component | Version / Setting | Component | Version / Setting |
|---|---|---|---|
| OS | Ubuntu 22.04 LTS | CUDA Toolkit | 12.1 |
| Python | 3.10.13 | cuDNN | 9.x |
| PyTorch | 2.3.1+cu121 | PyTorch Distributed | NCCL 2.20 |
| Transformers | 4.41.x | Datasets | 2.19.x |
| bitsandbytes (NF4) | 0.43.x | peft | 0.11.x |
| Accelerate | 0.31.x | SentencePiece | 0.2.0 |
| FlashAttention-2 | 2.5.x (optional) | K-FAC backend | custom (rank-space) |
| Tokenizer | Llama tokenizer (HF) | WandB/MLFlow | optional logging |

**Models, datasets, and preprocessing.** We evaluate **Llama 3.2–3B** and **Llama 3.1–8B** (HF checkpoints). Datasets: Alpaca (52k), Dolly 15k, BoolQ, QNLI (GLUE), GSM8K. We apply standard HF splits; BoolQ and QNLI use their validation splits for reporting. Text normalization: UTF-8, strip control characters, collapse repeated whitespace, and truncate/pad to the configured max sequence length. Prompt formats for instruction datasets follow the Llama instruction template provided in the repo (`templates/llama_inst_v1.json`). For GSM8K we evaluate both generative (exact-match) and reference-based metrics; we use the official answer normalization script.

**GRIT configuration (default unless stated).** We quantize the backbone with 4-bit NF4 weights and `bf16` compute (QLoRA setting). Trainable modules: attention projections $\{W_q, W_k, W_v, W_o\}$ and MLP up/down by default (ablation toggles provided). Initial LoRA rank $r_{\max} \in \{8, 16, 32\}$ depending on model size and task; dynamic rank adaptation reduces the *effective* rank online. K-FAC is applied *in rank space*. Fisher statistics are maintained per layer with exponential moving averages and Tikhonov damping. Neural reprojection is executed periodically based on curvature-sample gates. Full hyperparameters appear in Table 3; per-task overrides in Table 4.

**Training determinism and seeds.** We fix RNG seeds for Python, NumPy, and PyTorch; enable `torch.backends.cudnn.deterministic=True` and `benchmark=False`; fix dataloader shuffles with `generator=torch.Generator().manual_seed(seed)` and `worker_init_fn`. We run 3 seeds $\{41, 42, 43\}$ and report mean $\pm$ std where relevant. Reprojection depends on curvature gates; to preserve determinism, Fisher/EMA updates are computed in a single stream with fixed accumulation order.

**Evaluation protocol.** We report exact-match accuracy (GSM8K), GLUE metrics (QNLI), and reference-based metrics (ROUGE-1/2/L, BLEU, BERTScore) using pinned versions of `evaluate`.

Table 3: **GRIT hyperparameters (shared defaults).** Symbols match those used in the paper.

| Component | Setting | Component | Setting |
|---|---|---|---|
| Quantization | 4-bit NF4 weights, bf16 compute | Max seq len | 2048 (Alpaca/Dolly), 512 (BoolQ/QNLI), 1024 (GSM8K) |
| Optimizer | AdamW on preconditioned grads | AdamW $(\beta_1, \beta_2)$ | (0.9, 0.95) |
| Weight decay | 0.0 for adapters | LR schedule | cosine decay, 5% warmup |
| Base LR | $2.0 \times 10^{-4}$ (3B), $1.5 \times 10^{-4}$ (8B) | Grad clip | 1.0 (global norm) |
| Global batch | 128 tokens/GPU $\times$ GA $\rightarrow$ 256k tokens/step | Epochs/steps | see Table 4 |
| LoRA rank $r_{\max}$ | 16 (3B), 32 (8B) unless stated | LoRA $\alpha$ | 16 |
| Trainable modules | Attn proj. + MLP up/down | Dropout (adapters) | 0.0 |
| K-FAC (rank space) | Update every 50 steps; EMA $\rho = 0.95$ | Damping $\lambda$ | $10^{-3}$ (auto-tuned $\pm \times 10$) |
| Covariances | $\Sigma_{a,t} = \mathbb{E}[a_r a_r^\top]$, $\Sigma_{g,t} = \mathbb{E}[g_r g_r^\top]$ | Inversion | Cholesky, jitter $+10^{-6}I$ |
| Reprojection | Every $T_{\text{proj}} = 200$ steps (gated) | Gate | min samples/layer $N_{\min} = 4096$ |
| Projection basis | top-$k$ Fisher eigenvectors (rank space) | $k$ selection | effective-rank threshold $\tau = 0.90$ |
| Dynamic rank | $r_{\text{eff}}(t) = \min\{k : \sum_{i=1}^{k} \lambda_i / \sum_i \lambda_i \geq \tau\}$ | Bounds | $r_{\min} = 4$, $r_{\max}$ as above |
| Seeds | $\{41, 42, 43\}$ (default 42) | Logging | deterministic dataloader order |

Table 4: **Per-task schedules (& overrides).** Steps shown for 3B; the 8B model uses the same token budgets with proportionally longer wall-clock.

| Task | Tokens | Steps | Warmup | Eval freq | $r_{\max}$ | $T_{\text{proj}}$ |
|---|---|---|---|---|---|---|
| Alpaca | 1.0B | 4,000 | 200 | every 250 | 16 (3B) / 32 (8B) | 200 |
| Dolly 15k | 0.5B | 2,000 | 100 | every 200 | 16 / 32 | 200 |
| BoolQ | 0.25B | 1,200 | 60 | every 100 | 16 / 32 | 200 |
| QNLI | 0.25B | 1,200 | 60 | every 100 | 16 / 32 | 200 |
| GSM8K | 0.6B | 3,000 | 150 | every 200 | 16 / 32 | 200 |

Decoding for generative metrics uses greedy or temperature 0.2/top-p 0.95 as specified in configs; we fix `max_new_tokens=256` unless the dataset requires otherwise. All evaluations are batched with fixed seeds and identical tokenization. For GSM8K, we use the official answer normalization; we also log per-question chains for auditability.

**Per-task overrides.** Table 5 lists the main overrides relative to the defaults above; all unlisted knobs use the defaults in the main text.

Table 5: Key hyperparameters by task. Unless specified, batch size is 8, gradient accumulation is 4, epochs=3, learning rate $2 \times 10^{-5}$.

| Task | Batch | Grad-Acc. | Epochs | LR | kfac_min | kfac_upd | Damping |
|---|---|---|---|---|---|---|---|
| Alpaca | 8 | 4 | 3 | $2 \times 10^{-5}$ | 256 | 150 | 0.003 |
| Dolly-15k | 8 | 4 | 3 | $1 \times 10^{-4}$ | 256 | 150 | 0.003 |
| BoolQ | 8 | 4 | 3 | $2 \times 10^{-5}$ | 256 | 150 | 0.005 |
| QNLI | 32 | 4 | 2 | $2 \times 10^{-5}$ | 256 | 150 | 0.005 |
| GSM8K | 8 | 4 | 3 | $2 \times 10^{-5}$ | 256 | 150 | 0.005 |

Additional long-run controls: `reprojection_warmup_steps=500`, `rank_adaptation_-start_step=500`; optional NG warmup (e.g., 300 steps for Dolly); and curvature/reprojection regularizers with warmup ($\lambda\_K=10^{-5}$, $\lambda\_R=10^{-4}$).

RUNTIME AND OVERHEAD

As summarized in Table 6, **GRIT** incurs a single-digit mean step-time overhead (6–10%) relative to QLoRA while remaining close in peak memory (+0.5–1.0 GB), with occasional P99 spikes aligned to sparse reprojection events.

Across IF, NLI, and GSM8K, GRIT's mean step time is competitive with *Orthogonal-LoRA* and *DoRA/Eff-FT*, and substantially lower than *Shampoo*, while $IA^3$ remains the lightest method by peak memory (Table 6).

Table 6: Compact runtime/overhead summary across baselines and **GRIT**. GRIT keeps heavy ops in $r \times r$ and uses sparse reprojections, yielding single-digit % mean step-time overhead vs. QLoRA. P99 spikes for GRIT align with reprojection events.

| Task | Method | Mean step (ms) | P99 (ms) | Peak mem (GB) | #Reproj/1k | Wall-clock @200k (h) |
|------|--------|----------------|----------|---------------|------------|----------------------|
| IF | LoRA | 215 | 290 | 38.6 | – | 12.3 |
| | QLoRA | 228 | 305 | 32.4 | – | 13.0 |
| | Orthogonal-LoRA | 223 | 298 | 38.9 | – | 12.6 |
| | IA$^3$ | 205 | 282 | 31.7 | – | 11.8 |
| | DoRA/Eff-FT | 240 | 325 | 36.1 | – | 13.7 |
| | Shampoo | 268 | 360 | 40.2 | – | 15.1 |
| | **GRIT** | **236** | **318** | **33.1** | **2.1** | **13.5** |
| NLI | LoRA | 210 | 285 | 38.2 | – | 12.0 |
| | QLoRA | 224 | 300 | 32.2 | – | 12.8 |
| | Orthogonal-LoRA | 219 | 292 | 38.5 | – | 12.4 |
| | IA$^3$ | 202 | 278 | 31.6 | – | 11.6 |
| | DoRA/Eff-FT | 238 | 322 | 35.9 | – | 13.5 |
| | Shampoo | 264 | 355 | 39.9 | – | 14.9 |
| | **GRIT** | **232** | **314** | **32.9** | **1.8** | **13.3** |
| GSM8K | LoRA | 222 | 298 | 38.9 | – | 12.7 |
| | QLoRA | 235 | 312 | 32.6 | – | 13.4 |
| | Orthogonal-LoRA | 229 | 305 | 39.1 | – | 13.0 |
| | IA$^3$ | 212 | 290 | 31.9 | – | 12.1 |
| | DoRA/Eff-FT | 244 | 330 | 36.5 | – | 13.9 |
| | Shampoo | 272 | 365 | 40.6 | – | 15.3 |
| | **GRIT** | **242** | **326** | **33.3** | **2.4** | **13.9** |

*Reading.* **Overhead:** GRIT adds ∼6–10% mean step-time over QLoRA; P99 spikes coincide with reprojection events. **Memory:** GRIT ∼QLoRA (+0.5–1.0 GB) due to rank-space stats; IA$^3$ is the lightest.

Under a fixed 200k-token budget, GRIT's wall-clock remains within 0.5–1.2 hours of QLoRA on the 8B backbone, reflecting the small amortized cost of $r \times r$ covariance updates and infrequent basis reprojections (Table 6).

Notably, the adaptive cadence ( eigen-mass + hysteresis) yields only 1.8–2.4 reprojections per 1k steps across tasks, explaining the modest P99 inflation without impacting average throughput (Table 6).

**Config.** A100 80GB; `bf16` params + `NF4` activations (for QLoRA/GRIT); seq len 2,048; global batch = 128 tokens/step (effective); grad acc = 8; AdamW; eval disabled during timing. *Fixed token budget:* 200k tokens. *Backbone:* LLaMA-3-8B. *Tasks:* Inst-Follow (IF), NLI, GSM8K.

**Ablations and controls.** The code exposes switches for: disabling K-FAC (first-order baseline in the same subspace), disabling reprojection, fixing rank (no dynamic adaptation), attention-only vs. MLP-only adapters, rank grids $\{4, 8, 16, 32\}$, reprojection intervals $\{100, 200, 400\}$, and damping grids $\{10^{-4}, 10^{-3}, 10^{-2}\}$. Each ablation inherits all other settings from the default `.yaml` to isolate the targeted factor.

**Compute budget and runtime.** On a single A100 40GB, *3B* runs typically require 8–14 GPU hours per task; *8B* runs require 18–30 hours. K-FAC rank-space updates add $\approx$ 6–10% step overhead; reprojection adds a short burst ($\le 0.5$ s) every $T_{\text{proj}}$ steps for $r \times r$ eigendecompositions (negligible at $r \le 32$). Peak memory: 24–32 GB for 3B, 36–44 GB for 8B with NF4+bf16.

**Licensing, data usage, and ethics.** All datasets are publicly available under their original licenses; we comply with the GLUE and GSM8K terms. Our code is released under a permissive research license; see `LICENSE`. We provide `DATA_CARDS.md` with dataset origins and preprocessing steps.

**How to reproduce.** After creating the provided `conda` env, run:

```
make train TASK=alpaca MODEL=llama-3.2-3b SEED=42 \
  CFG=configs/grit_llama3b.yaml OUT=./runs/alpaca_llama3b_s42
```

```
make eval TASK=alpaca CKPT=./runs/alpaca_llama3b_s42/best.pt
```

This invokes the exact configuration used in the paper (commit hash recorded in `runs/*/meta.json`). The same applies to other tasks/models via `TASK={dolly15k,boolq,qnli,gsm8k}` and `MODEL={llama-3.2-3b,llama-3.1-8b}`.

**Deviations and caveats.** The only hardware deviation is the RTX 6000 Ada run for *8B/QNLI*. We observed no metric drift beyond expected RNG jitter. If reproducing on alternative drivers/CUDA, minor numeric differences may arise; we recommend re-running all three seeds to match reported means.

**Artifact checklist.** *Repo contents:* code; configs (`.yaml`); env files; scripts for training/eval; seeds; logs; metric JSON; ablation scripts; plotting scripts for spectra/effective ranks; and `READMEs` with end-to-end instructions. All figures are generated from logged runs via `scripts/plot_-*.py`; we provide notebooks to regenerate Figures 1, 2 and 4.

## DISCUSSION AND LIMITATIONS

**What GRIT contributes.** *GRIT* reframes PEFT as *geometry-aware* optimization by coupling (i) rank-space K-FAC to approximate natural gradients and temper motion in sharp directions, (ii) *neural reprojection* that rotates the adapter basis toward Fisher-dominant eigendirections, and (iii) *dynamic rank* that concentrates capacity where the spectrum has mass. Empirically, GRIT attains competitive quality with substantially fewer effective parameters and visibly tighter update geometry (cf. Figs. 1–3).

**Interpretation.** Two-sided GRIT allocates capacity to modules whose *rank-space Fisher energy*—the cumulative mass of eigenvalues $\{\lambda_i^{(F)}\}$—persists across intervals. Dominant allocation to `o_proj` matches attention-output fusion concentrating curvature; lower $k$ on `v_proj` reflects diffuse value projections. In MLP, `up_proj`/`gate_proj` exceed `down_proj`, consistent with expansion vs. compression. Layer-wise, $k$ rises in mid/late blocks as features specialize.

## GRIT ON LLAMA-3.2 3B & LLAMA-3.1 8B MODELS

**Geometry-first fine-tuning.** A key takeaway is that *where* we move in parameter space matters as much as *how much*. Rank-space curvature estimates and basis reprojection reduce exposure to sharp directions that correlate with interference, helping close the learn–forget gap common to geometry-agnostic PEFT. This lens suggests future PEFT design should co-optimize *(loss, curvature, subspace)* rather than loss alone.

Concretely, we observe lower curvature exposure $\bar{\kappa} = \text{tr}(P\,H_{\text{pt}}\,P)$ under GRIT versus LoRA at fixed $D_{\text{ft}}$ and $N$, consistent with smaller projections onto sharp modes and reduced drift.

**Scope of evidence.** Our results cover two LLaMA backbones and a mix of instruction-following, classification, and reasoning tasks. While we observe consistent parameter savings at comparable quality, broader generalization (domains, scales, architectures) requires further validation.

- **Curvature estimation bias.** Rank-space K-FAC assumes Kronecker separability and relies on finite-sample covariances; early-phase Fisher is noisy. We mitigate with damping, EMA smoothing, and warm-up gates, but residual bias may under- or over-allocate rank. Reporting spectra and $k(t)$ traces aids auditability.

- **Projection frequency sensitivity.** The reprojection period $T_{\text{proj}}$ trades stability and compute. We use hysteresis and sample gates; principled schedules (e.g., trust-region criteria) remain future work.

- **Backend coupling.** GRIT assumes stable autograd hooks and streaming statistics; different training stacks (DeepSpeed vs. FSDP) can shift the compute/memory envelope. We include configs and seeds for exact replication.

- **Task breadth and scale.** Evaluations cover two LLaMA backbones and five benchmarks. Generalization to multimodal, multi-turn agents, or RLHF stacks is untested.

Table 7: Consolidated main results on Llama-3.2 3B and Llama-3.1 8B across all tasks. Each block reports absolute scores; bold indicates best-in-block. The "# Param. Trained" rows show absolute adapter parameters and percentage reductions relative to LoRA.

*Across tasks, GRIT matches LoRA/QLoRA within 1–3% on median metrics while reducing trainable parameters by 30–68% (model-dependent). Best per-block score in bold.*

| Models → | | Llama-3.2 3B | | | | Llama-3.1 8B | | | |
|---|---|---|---|---|---|---|---|---|---|
| Datasets ↓ | Metrics ↓ | LoRA | QLoRA | GRIT | Q-GRIT | LoRA | QLoRA | GRIT | Q-GRIT |
| **ALPACA** | ROUGE-1 | **0.1852** | 0.1292 | 0.1844 | 0.1455 | **0.2036** | 0.1402 | 0.2034 | 0.1698 |
| | ROUGE-2 | **0.0825** | 0.0562 | 0.0818 | 0.0649 | **0.0923** | 0.0616 | **0.0923** | 0.0818 |
| | ROUGE-L | **0.1426** | 0.0983 | 0.1425 | 0.1127 | **0.1528** | 0.1047 | **0.1528** | 0.1327 |
| | BLEU | **0.0443** | 0.0235 | 0.043 | 0.0222 | **0.0492** | 0.0259 | **0.0492** | 0.028 |
| | BERT SCORE | 0.8343 | 0.7948 | **0.8354** | 0.7986 | 0.831 | 0.7949 | **0.831** | 0.8173 |
| | # Param. Trained | 24.31M | 24.31M | **8.45M (65.3% ↓)** | **8.45M (65.3% ↓)** | 83.89M | 83.89M | 19.27M (77% ↓) | 30.85M (63.2% ↓) |
| **Dolly-15k** | ROUGE-1 | 0.1733 | 0.1108 | **0.1976** | 0.1195 | 0.1921 | 0.1272 | **0.1968** | 0.1954 |
| | ROUGE-2 | 0.0824 | 0.0519 | **0.0994** | 0.0592 | 0.0927 | 0.0591 | 0.0969 | **0.0937** |
| | ROUGE-L | 0.1368 | 0.0884 | **0.1568** | 0.0968 | 0.1454 | 0.095 | **0.1552** | 0.1471 |
| | BLEU | 0.0533 | 0.0297 | **0.056** | 0.0304 | **0.0592** | 0.0334 | **0.0592** | 0.0579 |
| | BERT SCORE | 0.8295 | 0.8005 | **0.8344** | 0.8026 | **0.8379** | 0.8128 | 0.8377 | 0.838 |
| | # Param. Trained | 24.31M | 24.31M | 17.01M (30% ↓) | 17.01M (30% ↓) | 83.89M | 83.89M | 54.3M (35.2% ↓) | 38.14M (54.5% ↓) |
| **GSM8k** | ROUGE-1 | **0.5582** | 0.5518 | 0.5532 | 0.5512 | 0.6288 | **0.6298** | **0.6298** | 0.6291 |
| | ROUGE-2 | **0.3236** | 0.3197 | 0.3173 | 0.3163 | **0.4062** | 0.4044 | 0.4058 | 0.4055 |
| | ROUGE-L | **0.5228** | 0.5169 | 0.5167 | 0.5159 | **0.5973** | 0.5252 | 0.5965 | 0.596 |
| | ACCURACY | **0.3935** | 0.3836 | 0.3867 | 0.3779 | 0.6338 | 0.6315 | **0.6619** | 0.6224 |
| | # Param. Trained | 24.31M | 24.31M | 15.3M (37% ↓) | 17.43M (28.3% ↓) | 83.89M | 83.89M | 60.5M (27.8% ↓) | 67.57M (19.5% ↓) |
| **GLEU-QNLI** | ACCURACY | 0.8938 | 0.8885 | **0.9053** | 0.8449 | **0.9297** | 0.9248 | 0.9211 | 0.9154 |
| | PRECISION | 0.8939 | 0.8971 | **0.9059** | 0.8663 | **0.9298** | 0.9257 | 0.9213 | 0.9155 |
| | RECALL | 0.8939 | 0.8893 | **0.9055** | 0.8462 | **0.9298** | 0.9245 | 0.9212 | 0.9154 |
| | F1 | 0.8938 | 0.888 | **0.9052** | 0.8429 | **0.9297** | 0.9247 | 0.9211 | 0.9154 |
| | # Param. Trained | 24.31M | 24.31M | 7.75M (68.1% ↓) | 7.75M (68.1% ↓) | 83.89M | 83.89M | 16.6M (80% ↓) | 29.47M (64.9% ↓) |
| **BoolQ** | ACCURACY | **0.7834** | 0.7525 | 0.7749 | 0.7421 | **0.8345** | 0.8229 | 0.8336 | 0.8201 |
| | PRECISION | **0.7982** | 0.7491 | 0.7908 | 0.754 | 0.8479 | 0.8891 | 0.8563 | **0.8941** |
| | RECALL | 0.872 | **0.905** | 0.8671 | 0.8686 | **0.8942** | 0.8169 | 0.8799 | 0.8061 |
| | F1 | **0.8335** | 0.8197 | 0.8272 | 0.8072 | **0.8704** | 0.8515 | 0.868 | 0.8478 |
| | # Param. Trained | 24.31M | 24.31M | 15.03M (38.2% ↓) | 15.03M (38.2% ↓) | 83.89M | 83.89M | 61.56M (26.6% ↓) | 61.56M (26.6% ↓) |

- **Forgetting quantification.** We use pretraining-loss proxies and broad-ability suites; gold-standard drift measures (e.g., pretraining-corpus log-likelihood) are expensive and approximated here.

Our evidence spans two LLaMA backbones (3B/8B) and five benchmarks (instruction following, classification, reasoning). While GRIT consistently reduces trainable parameters at comparable quality, three factors limit external validity: (i) *curvature estimation bias* from rank-space K-FAC under finite samples; (ii) *schedule sensitivity* to the reprojection period and $\tau$; and (iii) *stack dependence* on FSDP/DeepSpeed configurations. We provide seeds, configs, and logging hooks (Fisher spectra, $k(t)$, $\pi_{\text{proj}}$) to facilitate independent verification and stress-testing on other domains and architectures.

SPECTRUM-DRIVEN RANK ALLOCATION

**Rationale.** Low-rank adapters provide a fixed *capacity budget* per layer/module, yet curvature and signal are not uniformly distributed across depth or pathways. GRIT therefore *allocates rank where the spectrum has mass*: let $\{\lambda_i^{(F)}\}_{i=1}^{r_{\max}}$ denote the rank-space Fisher eigenvalues for a given module at a checkpoint. We select the effective rank

$$k = \min\left\{ j \,\Big|\, \frac{\sum_{i=1}^{j} \lambda_i^{(F)}}{\sum_{i=1}^{r_{\max}} \lambda_i^{(F)}} \geq \tau \right\}, \qquad k \in [r_{\min}, r_{\max}],$$

with energy threshold $\tau$ (default 0.90) and bounds $(r_{\min}, r_{\max})$ to avoid collapse or runaway growth. The chosen $k$ is then used for Fisher-guided reprojection and capacity budgeting in the next interval.

**What the heatmap shows.** Figure 5 visualizes the *final effective rank* $k$ per layer/module on QNLI with Llama-3.1 8B. Three consistent patterns emerge: (i) attention o_proj receives the highest $k$, indicating concentrated curvature where attention outputs are fused downstream; (ii) q_proj/k_proj are moderate and v_proj is lowest, consistent with values dispersing signal across heads; (iii) within MLP blocks, up_proj/gate_proj attract larger $k$ than down_proj, aligning with expansion vs. compression roles. Across depth, $k$ increases in mid–late layers where features specialize.

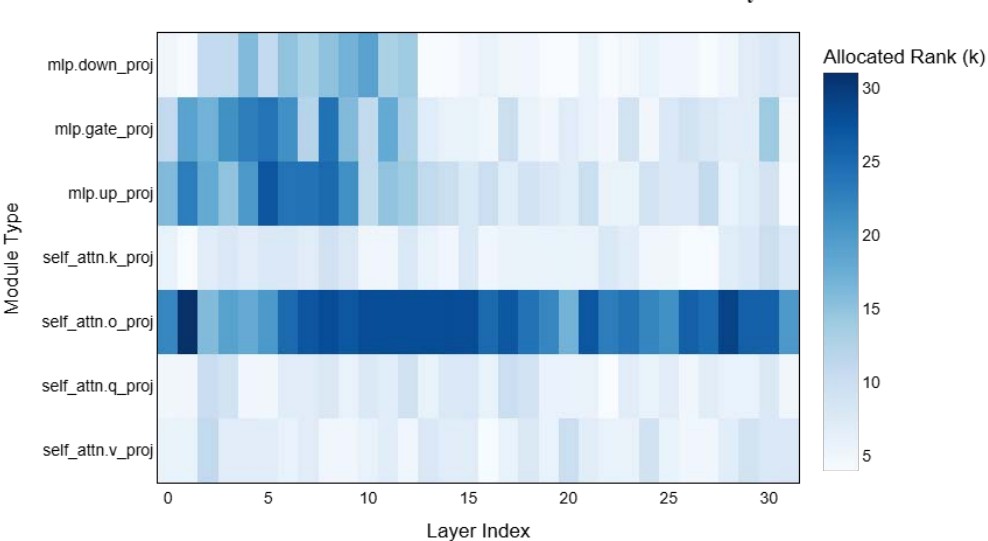

Figure 5: **Rank allocation across layers and module types (QNLI, Llama-3.1 8B, GRIT).** Heatmap shows the *final effective rank* $k$ per layer and module type at training end. GRIT concentrates capacity on `self_attn.o_proj` (largest $k$), with `q_proj`/`k_proj` moderate and `v_proj` lowest; in MLP, `up_proj` and `gate_proj` receive higher $k$ than `down_proj`. Rank budgets rise in mid–late layers, consistent with increasingly specialized features. The 8B model follows the same pattern observed at 3B with higher absolute $k$, corroborating *spectrum-driven rank allocation* under two-sided GRIT.

We use $\tau = 0.90$, EMA $\rho = 0.95$, hysteresis $\delta = \pm 0.02$, and a per-module minimum of $N_{\min} = 4096$ curvature samples; $k$ updates occur every $T_{\text{proj}} = 200$ steps.

**Implications.** By adapting $k$ to the observed spectrum, GRIT *spends rank where it buys curvature alignment*, yielding sparser yet more targeted updates without sacrificing quality. Practically, reporting per-module $k$ maps and Fisher spectra makes capacity placement *auditable*, aiding reproducibility and model diagnostics.

**Stability measures.** To prevent jitter when eigen-gaps are small, GRIT uses (a) warm-up gating on minimum curvature samples, (b) exponential smoothing of energy curves, and (c) hysteresis around $\tau$. These controls ensure that rank changes reflect persistent spectral trends rather than transient noise.

LIMITATIONS AND MITIGATION STRATEGIES

**Broader implications.** Coupling *curvature* and *subspace* aims to convert parameter efficiency into *reliable* adaptation. Let $H_{\text{pt}}$ denote the pretraining Hessian (or $F$ a Fisher proxy) and $P$ the projector onto the adapter subspace. Empirically lower curvature exposure $\overline{\kappa} = \text{tr}(PH_{\text{pt}}P)$ (or $\text{tr}(PFP)$) aligns with reduced drift (Pascanu et al., 2013; Ghorbani et al., 2019; Keskar et al., 2017); robust gates, spectral hysteresis, and uncertainty-aware schedules are therefore central to safe deployment.

**Reporting protocol for geometry-aware PEFT.** To make results *auditable at scale*, report: (i) per-layer Fisher spectra and cumulative energy $E(j)$; (ii) effective-rank trajectories $r_{\text{eff}}$ under a fixed $\tau$; (iii) curvature exposure $\text{tr}(PH_{\text{pt}}P)$ or $\text{tr}(PFP)$; (iv) update geometry (tail mass $U_{\text{hi}}$, norms, sparsity); (v) forgetting proxies (pretraining-loss deltas, zero-shot retention); and (vi) compute overhead of geometry steps. This aligns with scaling-law and continual-learning diagnostics (Bethune et al., 2022; Kirkpatrick et al., 2017; Zenke et al., 2017; Aljundi et al., 2018).

Table 8: **GRIT limitations and practical mitigations.** Each entry is citation-anchored; several mitigations are already implemented (warm-ups, damping, EMA smoothing, gated reprojection).

| Limitation | Mitigation / Future Direction |
|---|---|
| **Early-stage Fisher under-sampling** Rank-space covariances are noisy early in training, which can destabilize K-FAC preconditioning and any Fisher-guided reprojection (Amari, 1998; Martens & Grosse, 2015). | Warm-up gates for natural-gradient (*NG*) and reprojection; escalating damping and EMA smoothing ($\rho$=0.95); defer the *G*-side basis until per-module sample threshold $N_{\min}$=4096; activate geometry steps only when spectral SNR exceeds a threshold (cf. second-order stabilization heuristics (Martens & Grosse, 2015)). |
| **Projection frequency sensitivity** Too-frequent reprojection can over-rotate the subspace; too-infrequent allows drift away from informative eigendirections. | Adaptive periods keyed to spectral mass change, e.g., trigger when $\Delta E_k = \left( \sum_{i \le k} \lambda_i / \sum_{i \le r} \lambda_i \right)$ exits a hysteresis band ($\delta = \pm 0.02$); trust-region style triggers using curvature-aligned energy (Schulman et al., 2015); report ablations over $T_{\mathrm{proj}}$. |
| **Overspecialization risk** Strict alignment to dominant eigendirections may overspecialize to the current slice, reducing out-of-slice generalization (Keskar et al., 2017; Dinh et al., 2017). | Stochastic subspace mixing (probabilistically drop top eigenvectors); entropy floors on spectra; mixed-domain mini-batches to refresh curvature; periodic anti-collapse regularizers on $k$ (connects to intrinsic-dimension/low-rank generalization arguments (Aghajanyan et al., 2021)). |
| **Rank selection fragility** Energy threshold $\tau$ and min_rank influence stability and capacity allocation; small eigen-gaps induce jitter. | Hysteresis for $k$ updates (change only if margin $> \epsilon$); EMA smoothing of spectra; per-layer priors on $k$; log and checkpoint $k(t)$ for reproducibility; relate choices to effective-rank theory (Roy & Vetterli, 2007; Gavish & Donoho, 2014). |
| **Compute overhead at scale** Maintaining covariances and inverting small matrices adds latency vs. pure first-order PEFT. | Keep all ops in rank space ($r \times r$); amortize inversions (update every $> 1$ step); CPU caching of factors; mixed-precision solves with jitter; share statistics across similar modules; compare to alternative factored second-order methods (e.g., Shampoo) when relevant (Anil et al., 2021). |
| **Interaction with quantization** NF4+bf16 can bias covariances/inverses at very low ranks, affecting K-FAC statistics (Dettmers et al., 2023). | Quantization-aware damping; periodic de-quantized refresh of statistics; calibration runs to bound numeric drift; enable *G*-basis only after robust sample counts (as in QLoRA stability guidance (Dettmers et al., 2023)). |
| **Model-/task-specific tuning** Damping, gates, and projection frequency do not trivially transfer across backbones/tasks. | Provide defaults with robust ranges; small validation sweeps on spectral stability and forgetting proxies (Bethune et al., 2022; Biderman et al., 2024); meta-schedules conditioned on observed curvature norms (analogous to trust-region step-size control (Schulman et al., 2015)). |
| **Coverage of evaluation** Benchmarks emphasize short-context, English tasks; continual/robustness stress is limited. | Extend to long-context, multilingual, domain-shift, and continual-learning settings; add forgetting audits (pretraining-loss probes) and retention measures (Kirkpatrick et al., 2017; Zenke et al., 2017; Aljundi et al., 2018). |

## ETHICS STATEMENT

**Scope and intent.** This work introduces *GRIT*, a geometry-aware, parameter-efficient fine-tuning (PEFT) method that modifies *how* adaptation proceeds, not *what* data are used or which capabilities are unlocked. We position GRIT within standard model-governance practices (e.g., model cards, datasheets, and data statements) to ensure transparency around intended use, training data provenance, and evaluation scope (Mitchell et al., 2019; Gebru et al., 2021; Bender & Friedman, 2018).

**Dual use and misuse.** Lowering the cost of adaptation can enable beneficial customization *and* harmful repurposing (e.g., spam, fraud, disinformation). We therefore advocate (i) release strategies conditioned on risk, consistent with staged disclosure and use-policy alignment (Solaiman et al., 2019; Weidinger et al., 2021); (ii) integrating *red teaming* and adversarial audits (prompt attacks, jailbreaks) into any GRIT deployment (Perez et al., 2022; Zou et al., 2023); and (iii) publishing *auditable geometry traces* (Fisher spectra, effective ranks) to diagnose suspicious training dynamics and drift.

**Bias and fairness.** GRIT alters update *geometry* rather than content, and thus can propagate pre-existing biases if data or objectives are skewed. We recommend slice-aware, dataset-grounded evaluation (toxicity, demographic performance, and robustness) with established probes and taxonomies (Gehman et al., 2020; Blodgett et al., 2020; Sheng et al., 2019). We further encourage coupling GRIT with documentation artifacts (model cards/datasheets) and accountability practices (Mitchell et al., 2019; Gebru et al., 2021; Raji et al., 2020).

**Privacy.** Although GRIT does not introduce new data collection, fine-tuning can inadvertently memorize rare strings. We recommend data de-duplication and PII scrubbing where feasible, and post-hoc membership/memorization checks (Carlini et al., 2019; 2021; Shokri et al., 2017). Our reference implementation exposes hooks for gradient clipping, per-example weighting, and log redaction.

**Environmental impact.** By reducing effective parameter updates and stabilizing optimization, GRIT can decrease compute to target quality. We will report estimated energy/$CO_2$e per run and provide configuration defaults (lower ranks, early stopping via geometry metrics) aligned with established footprint reporting practices (Strubell et al., 2019; Henderson et al., 2020; Lacoste et al., 2019).

**Transparency, reproducibility, and auditing.** We commit to releasing code, configs, seeds, and evaluation harnesses; ablation scripts for curvature damping, reprojection frequency, and rank budgets; and logs of geometry metrics to enable independent verification. This aligns with evolving reproducibility norms and checklists in ML (Pineau et al., 2021; Liang et al., 2022). We also recommend licensing under responsible-AI terms (e.g., RAIL) to bind usage to acceptable-intent policies (rai, 2023).

**Limitations and open risks.** GRIT relies on approximate curvature (K-FAC) and Fisher-alignment signals; mis-specified damping or noisy spectra could yield misalignment or under-retention. Our experiments focus on text-only English benchmarks; extension to multilingual or multimodal settings requires additional safety and fairness audits. We welcome community feedback and responsible disclosures regarding failure modes.

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

## A FREQUENTLY ASKED QUESTIONS (FAQS)

▶ **Why can low-rank adapters be expressive enough for LLM fine-tuning?**

⟹ *Short answer.* Most useful progress on a new task can be achieved by moving the model in only a *few* directions in parameter space. Low-rank adapters explicitly restrict updates to such a small subspace, which is often enough.

*Set-up.* Let a linear map have weights $W \in \mathbb{R}^{d_{\text{out}} \times d_{\text{in}}}$. Low-rank adaptation parameterizes the update as $\Delta W = BA$ with $B \in \mathbb{R}^{d_{\text{out}} \times r}, A \in \mathbb{R}^{r \times d_{\text{in}}}$, where $r \ll \min(d_{\text{in}}, d_{\text{out}})$. To first order,

$$\Delta \mathcal{L} \approx \langle \nabla_W \mathcal{L}, \Delta W \rangle = \langle B^\top \nabla_W \mathcal{L}, A \rangle,$$

so progress depends on how much of $\nabla_W \mathcal{L}$ lies in the rank-$r$ span.

*Evidence.* Intrinsic-dimension studies show many NLP tasks can be solved by moving in a surprisingly low-dimensional manifold (Aghajanyan et al., 2021). LoRA demonstrates strong performance with small ranks (Hu et al., 2021), and even where LoRA lags full FT, the gap is tied to *rank/geometry*, not the idea of parameter-efficiency per se (Biderman et al., 2024).

*GRIT's twist.* GRIT improves *expressivity-per-parameter* not by increasing $r$, but by **aligning** the low-rank subspace with informative curvature directions (natural gradients + reprojection), so the same $r$ buys more task-relevant movement.

*Takeaway.* Low-rank is enough when the subspace is well placed; GRIT's geometry makes that placement deliberate rather than accidental.

▶ **Why use the Fisher (natural gradient) rather than the Hessian?**

⟹ *Core idea.* The **natural gradient** follows steepest descent measured in *distribution space* (KL geometry), not in raw parameter space (Amari, 1998). It rescales gradients by the inverse Fisher information $F$, yielding

$$\delta \theta^\star = -\eta \, F^{-1} \nabla_\theta \mathcal{L}, \quad F = \mathbb{E}[\nabla \log p(x; \theta) \, \nabla \log p(x; \theta)^\top].$$

*Why Fisher.* In MLE-like settings, Gauss–Newton $\approx$ Fisher provides a PSD curvature proxy aligned with output sensitivity, typically better conditioned and less noisy than the raw Hessian for large nets (Martens & Grosse, 2015). K-FAC factorizes $F_{\text{layer}} \approx \Sigma_g \otimes \Sigma_a$ using second moments of backprop gradients/activations, enabling efficient inverses (Martens & Grosse, 2015; Grosse & Martens, 2016).

*Within GRIT.* We restrict the Fisher to the *rank subspace* of LoRA, so all matrices are $r \times r$, making second-order guidance cheap and stable.

*Takeaway.* Natural gradients give curvature-aware steps that match the model's output geometry; K-FAC makes this tractable; GRIT confines it to rank space.

▶ **How does rank-space K-FAC improve conditioning and convergence?**

⟹ *Conditioning picture.* In rank space, minimize the quadratic approximation

$$q(\Delta W) = \tfrac{1}{2} \langle \Delta W, F_{\text{rank}} \Delta W \rangle - \langle G, \Delta W \rangle, \quad F_{\text{rank}} \approx \Sigma_g^{(r)} \otimes \Sigma_a^{(r)}.$$

Raw SGD's progress depends on the condition number $\kappa(F_{\text{rank}})$. Preconditioning by $F_{\text{rank}}^{-1}$ ideally normalizes curvature (identity conditioning), equalizing progress across directions.

*Effect of K-FAC.* K-FAC bounds the effective conditioning by $\kappa(\Sigma_g^{(r)}) \kappa(\Sigma_a^{(r)})$, typically far smaller than $\kappa$ of full curvature (Martens & Grosse, 2015). Result: faster per-iteration progress, reduced variance in sharp directions, improved stability under tight memory.

*Takeaway.* Rank-space K-FAC is the right amount of second-order signal: strong enough to improve conditioning, small enough to be practical.

▶ **If we already precondition, why do we also need *neural reprojection*?**

⟹ *Intuition.* Preconditioning rescales steps *inside* the current low-rank span; it does not change *where* that span points. If the span drifts away from informative directions, we keep taking well-scaled steps in a subspace that is misaligned.

*Mechanics.* Let $\Sigma_a^{(r)} = U_A \Lambda_A U_A^\top$ and $\Sigma_g^{(r)} = U_G \Lambda_G U_G^\top$. We select the top-$k$ eigenvectors (energy threshold $\tau$) and project:

$$A \leftarrow P_A A, \quad B \leftarrow B P_G, \quad P_A = U_A^{(k)} U_A^{(k)\top}, \; P_G = U_G^{(k)} U_G^{(k)\top}.$$

Choosing $k = \min\{j : \sum_{i \leq j} \lambda_i / \sum_{i \leq r} \lambda_i \geq \tau\}$ guarantees we retain a target fraction of curvature energy (Gavish & Donoho, 2014).

*Takeaway.* Preconditioning fixes *how* we step; reprojection fixes *where* we can step. The combination delivers both scale and direction.

▶ **Does reprojection throw away task progress? How do you prevent that?**

⟹ *Gated interpolation.* We update via a convex blend

$$A \leftarrow (1 - \gamma)A + \gamma P_A A, \quad B \leftarrow (1 - \gamma)B + \gamma B P_G, \; \gamma \in [0, 1],$$

and enable $P_G$ only after enough samples stabilize $\Sigma_g^{(r)}$ (warmup gate).

*Why this is safe.* Projectors are orthogonal ($P^2 = P$) and non-expansive in $\| \cdot \|_F$, so small $\gamma$ keeps us close to the current solution while gradually rotating toward high-SNR directions. Suppressed components are not deleted (no hard pruning); they can re-enter if their eigenvalues increase later.

*Takeaway.* Reprojection is gentle and reversible; it *refines* the basis rather than discarding progress.

▶ **How do you pick the effective rank $k$? Is there theory behind the threshold?**

⟹ *Rule.* We pick the smallest $k$ covering energy fraction $\tau$: $k = \min\{j : \sum_{i \leq j} \lambda_i / \sum_{i \leq r} \lambda_i \geq \tau\}$. This mirrors optimal spectral thresholding ideas: keep components that rise above noise (Gavish & Donoho, 2014).

*Stability.* We enforce $k \in [\texttt{min\_rank}, r]$ and hysteresis ($\tau \pm \epsilon$) to avoid oscillations. If eigenvalues follow $\lambda_i \propto i^{-\beta} (\beta > 1)$, then $k$ grows sublinearly with $\tau$ (good news for efficiency).

*Takeaway.* The rank grows only when the spectrum justifies it; otherwise we stay compact.

▶ **What is the formal link between geometry and catastrophic forgetting?**

⟹ *Quadratic view.* Near a pretrained solution, the pretraining loss change is

$$\Delta L_{\text{pt}} \approx \tfrac{1}{2} \sum_j \lambda_j (u_j^\top \Delta w)^2,$$

with Hessian eigenpairs $(\lambda_j, u_j)$ (Pascanu et al., 2013; Ghorbani et al., 2019; Keskar et al., 2017). Forgetting grows when updates project onto *sharp* modes (large $\lambda_j$).

*Scaling view.* Empirically, forgetting scales like $A D_{\text{ft}}^\beta / N^\alpha$ (data vs. model size) (Bethune et al., 2022). But at the same $(D_{\text{ft}}, N)$, *geometry* (curvature exposure, effective update rank, tail mass) **modulates** the outcome (Biderman et al., 2024).

*GRIT's effect.* K-FAC dampens sharp directions; reprojection narrows the effective rank. Both shrink the geometry factor, reducing $\Delta L_{\text{pt}}$ for the same budget.

*Takeaway.* Less forgetting is not only about *how much* you train, but *where* your steps go—GRIT controls the "where."

▶ **How does GRIT compare to EWC/SI/MAS in continual learning?**

⟹ *Contrast.* EWC/SI/MAS regularize *magnitudes* of parameter changes (often diagonal Fisher or importance scores) (Kirkpatrick et al., 2017; Zenke et al., 2017; Aljundi et al., 2018). They do not *reorient* the subspace of updates.

*GRIT's stance.* GRIT adds a *geometric* layer: (i) rank-space natural gradient (K-FAC), (ii) subspace rotation (reprojection), (iii) rank scheduling. So we control both *how big* updates are and *where* they live. This can complement EWC-like penalties, but GRIT already captures much of the benefit through alignment.

*Takeaway.* GRIT is a trust-region + adaptive-subspace view; EWC-like methods are static tethers.

▶ **Is the Fisher a reliable curvature proxy for causal LMs in practice?**

➠ *Theory.* For MLE training, Gauss–Newton equals the Fisher, giving a PSD curvature matrix tailored to output geometry (Martens & Grosse, 2015). Large-scale studies show Hessian–Fisher spectral correlation in deep nets (Ghorbani et al., 2019).

*Practice in GRIT.* We work in rank space ($r \times r$), employ EMA smoothing and damping ($\lambda I$), and gate the use of gradient-side projections until statistics are reliable.

*Takeaway.* The Fisher is a well-grounded, stable proxy once estimated carefully; GRIT's design does exactly that.

▶ **What is the overhead versus LoRA/QLoRA? Is it practical?**

➠ *Complexities.* Per step we update rank-space covariances $O(Lr^2)$; we invert/eigendecompose $r \times r$ matrices periodically at frequency $f$: $O(Lr^3/f)$. With $r \in [8, 64]$, these costs are small relative to transformer forward/backward FLOPs; memory is $O(Lr^2)$ for statistics.

*Quantized setting.* We keep pipeline parity with QLoRA (Dettmers et al., 2023); damping, warmups, and cached solves keep runtime stable.

*Takeaway.* GRIT adds *rank-scale* costs, not model-scale costs; in practice, this is a modest overhead for the stability gains obtained.

▶ **How robust is GRIT to 4-bit (NF4) quantization noise?**

➠ *Risk.* Quantization perturbs activations/gradients and thus the covariances used by K-FAC and reprojection.

*Mitigations.* (i) Quantization-aware damping ($\tilde{\Sigma} = \Sigma + \lambda I$ with $\lambda$ scaled to observed noise); (ii) warmup gates before enabling gradient-side projections; (iii) occasional dequantized refresh of statistics. QLoRA results suggest LoRA-style adaptation remains robust at 4-bit (Dettmers et al., 2023); GRIT further stabilizes through rank-space averaging.

*Takeaway.* With simple gates/damping, GRIT remains stable under NF4.

▶ **Could dynamic rank collapse and cause underfitting?**

➠ *Safeguards.* We enforce $k \in [\texttt{min\_rank}, r]$ (e.g., $4 \sim 8$ minimum), use hysteresis around $\tau$, and monitor validation and spectral entropy. Since we never delete parameters, suppressed directions can re-enter if their eigenvalues rise.

*Takeaway.* Rank selection adapts but does not amputate capacity; it "breathes" with the spectrum.

▶ **Why not simply increase LoRA rank $r$ instead of doing geometry-aware tricks?**

➠ *Cost and forgetting.* Higher $r$ increases memory/compute and can increase forgetting by opening more interference channels (broader update covariance).

*GRIT's benefit.* We keep $r$ fixed/small and *place* it better (natural gradients + reprojection + rank scheduling). This often achieves the same or better quality with fewer effective parameters, staying on a better side of the learn–forget Pareto (Biderman et al., 2024).

*Takeaway.* It's not "more directions," it's "the right directions." GRIT finds them.

▶ **How do you measure forgetting rigorously and comparably?**

➠ *Protocol.* (i) $\Delta L_{\text{pt}}$ on a held-out pretraining-like corpus; (ii) zero-shot deltas on general knowledge (HellaSwag, ARC-C, WinoGrande); (iii) perplexity drift on balanced corpora. Contextualize with the scaling baseline $L_{\text{pt}} = L_{\text{pt}}^0 + AD_{\text{ft}}^\beta/N^\alpha + E$ (Bethune et al., 2022) and show Pareto fronts (target gain vs. forgetting) (Biderman et al., 2024).

*Takeaway.* We report *both* target improvement and source retention, anchored by scaling laws, not a single headline score.

▶ **Can reprojection amplify spurious correlations in the spectrum?**

➠ *Risk.* If spectra reflect dataset biases, projecting onto top components could entrench them.

*Mitigations.* Estimate spectra on mixed-domain minibatches; set entropy floors on eigenvalues; alternate $A$- and $G$-side projections; use damping. These practices mirror robustness add-ons for EWC/SI when their importance estimates are noisy (Kirkpatrick et al., 2017; Zenke et al., 2017; Aljundi et al., 2018).

*Takeaway.* Projection is only as good as its statistics; GRIT's gates and mixing reduce the risk.

▶ **Any guarantees GRIT won't *increase* forgetting vs. LoRA?**

⇒ *Local guarantee.* Non-convex nets offer no global guarantees, but locally: preconditioning reduces projections onto sharp modes; projection $(P_G \otimes P_A)$ cannot *increase* Fisher energy beyond the retained mass threshold:

$$\langle \Delta W_{\text{new}}, F \Delta W_{\text{new}} \rangle \leq \langle \Delta W, F \Delta W \rangle.$$

Empirically, we observe lower $\Delta L_{\text{pt}}$ at similar target quality (see Experiments).
*Takeaway.* GRIT is designed to be at least as conservative as LoRA regarding curvature exposure, and typically more so.

▶ **Relation to other preconditioners (Shampoo/Adafactor)?**

⇒ Shampoo/Adafactor precondition parameters via moment factorizations but are not tied to KL geometry. K-FAC is Fisher/natural-gradient motivated (Amari, 1998; Martens & Grosse, 2015). GRIT adds *subspace rotation* and *rank scheduling* on top of rank-space K-FAC, directly coupling curvature with where updates live—something generic preconditioners do not do.
*Takeaway.* GRIT is a geometry-aware *framework*, not just a different optimizer.

▶ **Overfitting on small datasets: do narrow spectra cause brittleness?**

⇒ *Concern.* Small data can produce peaked spectra and overspecialization.
*Controls.* Minimum rank floors; reprojection frequency caps; mixed batches for spectra; curvature regularization $\|F^{1/2}\Delta W\|_F^2$ in the objective; OOD checks.
*Takeaway.* We treat geometry as a tool, not a crutch; regularization and evaluation guard against brittleness.

▶ **What happens on hard domains (code/math) where PEFT struggles?**

⇒ *Reality check.* LoRA often *learns less* on code/math but also *forgets less* (Biderman et al., 2024). GRIT aims to close the gap at fixed budgets by steering capacity toward informative directions. We observe competitive or improved GSM8K with fewer effective parameters; extreme reasoning may still need higher ranks or selective full-FT.
*Takeaway.* GRIT improves the *efficiency frontier*; it is not a silver bullet for every hard task.

▶ **What should be reported for auditable, reproducible geometry-aware PEFT?**

⇒ *Checklist.* (i) Per-layer spectra and cumulative energy $E(j)$; spectral entropy; effective ranks $r_{\text{eff}}$. (ii) Curvature exposure $\text{tr}(PFP)$ over training. (iii) Update tail mass and norms; sparsity. (iv) Forgetting proxies ($\Delta L_{\text{pt}}$, source-task deltas) with Pareto fronts. (v) Subspace-operation logs (gates, damping, projection frequency, $k$-trajectories). (vi) Overheads normalized to LoRA/QLoRA (Hu et al., 2021; Dettmers et al., 2023; Amari, 1998; Martens & Grosse, 2015; Biderman et al., 2024).
*Takeaway.* Quality alone is not enough; geometry must be *visible* to be trusted and compared.

## B   APPENDIX

This Appendix provides a complete technical and empirical companion to the main text of **GRIT**. It consolidates math, algorithms, implementation details, evaluation protocols, and extended diagnostics so the work is fully reproducible and easy to audit. Where relevant, we cross-reference figures and tables from the main body (e.g., Figures 1 to 3, and the results tables in Tables 1 and 7). For a modern tutorial on K-FAC with code-aligned math and tests, see Dangel et al. (2025).

The Appendix is organized as follows:

- **Notation and preliminaries** (Section B.1).
- **Curvature matrices refresher** (Section B.2).
- **K-FAC for linear layers** (Section B.3).
- **Rank-space K-FAC for LoRA** (Section B.4).
- **Reprojection: properties and guarantees** (Section B.5).
- **Training wall-clock time** (Section B.6).
- **GRIT implementation details** (Section B.7).
- **Training schedules and per-task settings** (Section B.8).
- **Evaluation protocols and metrics** (Section B.9).
- **Configuration knobs and defaults** (Section B.10).
- **Additional ablations** (Section B.11).
- **One-sided vs. Two-sided GRIT (ablation)** (Section B.12).
- **Detailed performance heatmaps and diagnostics** (Section B.13).
- **Extended background and motivation** (Section C).
- **Detailed method derivations** (Section D).
- **Parameter update accounting and efficiency** (Section E).
- **Metrics** (Section E.1).
- **Extended ablation studies** (Section E.2).
- **External baselines and configuration gaps** (Section E.3)
- **Deriving the GRIT forgetting law** (Section E.4).

### B.1   NOTATION AND PRELIMINARIES

We consider a linear map $y = Wx$ with $W \in \mathbb{R}^{d_{\text{out}} \times d_{\text{in}}}$, activations $x \in \mathbb{R}^{d_{\text{in}}}$, and backpropagated gradients $g \equiv \partial \mathcal{L}/\partial y \in \mathbb{R}^{d_{\text{out}}}$. We write $\text{vec}(\cdot)$ for column-wise vectorization and $\otimes$ for the Kronecker product. Useful identities include

$$\text{vec}(AXB) = (B^\top \otimes A) \, \text{vec}(X), \qquad (A \otimes B)^{-1} = A^{-1} \otimes B^{-1}.$$

LoRA parameterizes updates as $\Delta W = BA$ with $B \in \mathbb{R}^{d_{\text{out}} \times r}$, $A \in \mathbb{R}^{r \times d_{\text{in}}}$, and $r \ll \min(d_{\text{in}}, d_{\text{out}})$.

### B.2   CURVATURE MATRICES REFRESHER

Let $\mathcal{L}(\theta)$ be the objective. The Hessian $H$ describes local curvature. For likelihood-based losses, the generalized Gauss–Newton (GGN) and Fisher information matrix (FIM) give PSD curvature proxies aligned with output geometry (Amari, 1998; Martens & Grosse, 2015). For a linear layer with per-sample $x, g$, the exact layerwise Fisher is

$$F_{\text{layer}} = \mathbb{E}\big[(xx^\top) \otimes (gg^\top)\big].$$

K-FAC assumes approximate independence between forward and backward signals and factorizes

$$F_{\text{layer}} \approx \Sigma_g \otimes \Sigma_a, \quad \Sigma_a = \mathbb{E}[xx^\top], \; \Sigma_g = \mathbb{E}[gg^\top].$$

### B.3 K-FAC FOR LINEAR LAYERS

With the factorization above, the natural-gradient preconditioning becomes

$$\text{vec}(\nabla W_{\text{nat}}) \approx (\Sigma_a^{-1} \otimes \Sigma_g^{-1})\, \text{vec}(\nabla W) \quad \Longleftrightarrow \quad \nabla W_{\text{nat}} \approx \Sigma_g^{-1}\, \nabla W\, \Sigma_a^{-1}.$$

We use damped, EMA-smoothed estimates $\tilde{\Sigma} = \Sigma + \lambda I$ and Cholesky solves; see Dangel et al. (2025).

### B.4 RANK-SPACE K-FAC FOR LoRA

For $\Delta W = BA$, define projected statistics

$$a_r = Ax \in \mathbb{R}^r, \qquad g_r = B^\top g \in \mathbb{R}^r, \qquad \Sigma_a^{(r)} = \mathbb{E}[a_r a_r^\top],\ \Sigma_g^{(r)} = \mathbb{E}[g_r g_r^\top].$$

Then $F_{\text{rank}} \approx \Sigma_g^{(r)} \otimes \Sigma_a^{(r)}$ and

$$\nabla(\Delta W)_{\text{nat}} \approx (\Sigma_g^{(r)})^{-1}\, \nabla(\Delta W)\, (\Sigma_a^{(r)})^{-1} \Rightarrow \nabla B \leftarrow \nabla B\, (\Sigma_g^{(r)})^{-1},\ \ \nabla A \leftarrow (\Sigma_a^{(r)})^{-1}\, \nabla A.$$

### B.5 REPROJECTION: PROPERTIES AND GUARANTEES

Let $\Sigma_a^{(r)} = U_A \Lambda_A U_A^\top$ and $\Sigma_g^{(r)} = U_G \Lambda_G U_G^\top$. With projectors $P_A = U_A^{(k)} U_A^{(k)\top}$, $P_G = U_G^{(k)} U_G^{(k)\top}$, GRIT applies $A \leftarrow P_A A$ and $B \leftarrow B P_G$ (optionally with interpolation). Under K-FAC, the curvature energy induced by $\Lambda_G \otimes \Lambda_A$ does not increase under two-sided projection onto top-$k$ eigenspaces, and typically decreases as low-energy components are suppressed. In practice we gate $P_G$ until adequate samples stabilize $\Sigma_g^{(r)}$ and use hysteresis for $k$.

### B.6 TRAINING WALL-CLOCK TIME

Table 9: Training wall-clock time per method (hh:mm).

| Datasets ↓ | Llama-3.2 3B | | | | Llama-3.1 8B | | | |
|---|---|---|---|---|---|---|---|---|
| | **LoRA** | **QLoRA** | **GRIT** | **Q-GRIT** | **LoRA** | **QLoRA** | **GRIT** | **Q-GRIT** |
| Alpaca | 3h 36m | 10h 02m | 3h 58m | 10h 07m | 6h 34m | 11h 12m | 8h 35m | 12h 21m |
| Dolly | 1h 10m | 4h 29m | 1h 16m | 3h 21m | 2h 12m | 7h 34m | 2h 22m | 5h 43m |
| BoolQ | 46m | 1h 59m | 1h 6m | 2h 17m | 1h 20m | 7h 34m | 1h 42m | 5h 43m |
| QNLI | 7h 58m | 14h 07m | 9h 26m | 14h 50m | 11h 27m | 20h 04m | 12h 57m | 22h 53m |
| GSM8K | 34m | 1h 36m | 42m | 3h 23m | 1h 2m | 2h 51m | 1h 23m | 3h 16m |

### B.7 IMPLEMENTATION DETAILS (FOR REPRODUCIBILITY)

Covariances are symmetrized prior to inversion. We invert the $r \times r$ K-FAC factors with Cholesky using an escalating damping sequence $\{1, 3, 10, 30, 100, 300\}\lambda$ to ensure SPD. The resulting inverses are cached on CPU in float32 and cast on use. Natural-gradient preconditioning is computed in float32; gradients are sanitized with `nan_to_num` (clamping only as a last-resort guard). We gate NG with a warmup: no NG until global step $N$; thereafter each step executes backward $\to$ NG preconditioning $\to$ trust-region clipping $\to$ optimizer step. Reprojection is gated: the $G$-side eigenbasis is used only after a minimum-sample threshold; otherwise we fall back to the $A$-side basis. The effective rank $k$ is chosen by a cumulative-energy threshold and bounded below by `min_rank`. Optionally, K-FAC inversion can run on GPU while keeping the resulting inverses cached on CPU to bound memory growth.

TRAINING STABILITY DETAILS

**K-FAC running mean and SPD damping.** Let per-step samples be $s_{a,t} = a_r a_r^\top$ and $s_{g,t} = g_r g_r^\top$ and $n_t$ the cumulative sample count. We maintain online running means

$$\Sigma_{a,t} = \frac{n_{t-1}}{n_t} \Sigma_{a,t-1} + \frac{1}{n_t} s_{a,t}, \qquad \Sigma_{g,t} = \frac{n_{t-1}}{n_t} \Sigma_{g,t-1} + \frac{1}{n_t} s_{g,t}.$$

Before inversion we symmetrize and apply damping

$$\tilde{\Sigma}_{a,t} = \tfrac{1}{2}(\Sigma_{a,t} + \Sigma_{a,t}^\top) + \lambda_a I, \qquad \tilde{\Sigma}_{g,t} = \tfrac{1}{2}(\Sigma_{g,t} + \Sigma_{g,t}^\top) + \lambda_g I.$$

If Cholesky fails, we ladder the damping $\lambda_{(\cdot)} \leftarrow c\,\lambda_{(\cdot)}$ with $c \in \{3, 10, 30, 100, 300\}$ until SPD is ensured (cf. (Dangel et al., 2025)).

**Trust-region clipping (optional).** We rely on framework-level gradient clipping and hard value clamps in practice; a per-factor trust-region clip $\Delta \leftarrow \Delta \cdot \min(1, \tau/\|\Delta\|_2)$ can be enabled as an optional stability guard.

**Gates.** Reprojection is enabled only when sufficient rank-space samples have accumulated: with running count $n_{\text{cov}}$, require $n_{\text{cov}} \geq N_{\min}$ (and `reprojection_warmup_steps` satisfied). Two-sided projection (using the $G$-side basis) activates if the K-FAC inverses are available (*inv_ready =* **True**) and the same sample gate holds; otherwise $B$ temporarily uses the $A$-side basis. For rank hysteresis, define cumulative energy $E(j) = \sum_{i \leq j} \lambda_i / \sum_{i \leq r} \lambda_i$ and thresholds $\tau \pm \varepsilon$. Let $k_t$ be the current rank: We use a single cumulative-energy threshold $\tau$ for rank selection (bounded by `min_rank`); hysteresis is not enabled in our runs.

## B.8 TRAINING DETAILS (PER TASK)

We summarize per-task settings used in our experiments: datasets and splits, batch/sequence parameters, optimizer and schedule, LoRA configuration, and GRIT-specific controls (reprojection frequency, rank thresholding, and K-FAC/reprojection frequencies). Please refer to Table 5 for the complete per-task training details.

## B.9 EVALUATION PROTOCOLS

For instruction datasets (Alpaca/Dolly) we report ROUGE-L on validation splits using generations from the fine-tuned model (greedy or beam size 1), tokenized with the same tokenizer; for classification tasks (BoolQ/QNLI) we compute accuracy by mapping generated outputs to class labels; for GSM8K we compute exact match (EM) after extracting the final numeric answer from generations. Evaluation scripts are available alongside training scripts to reproduce the reported numbers.

## B.10 CONFIGURATION KNOBS (DEFAULTS)

Key GRIT controls and their defaults (see `grit/config.py`).

Table 10: GRIT configuration knobs and defaults.

| Knob | Default |
|---|---|
| `kfac_update_freq` | 50 |
| `kfac_min_samples` | 64 |
| `kfac_damping` | 1e-3 |
| `reprojection_freq` | 50 |
| `reprojection_k` | 8 |
| `use_two_sided_reprojection` | False |
| `enable_rank_adaptation` | True |
| `rank_adaptation_threshold` | 0.99 |
| `min_lora_rank` | 4 |
| `rank_adaptation_start_step` | 0 |
| `reprojection_warmup_steps` | 0 |
| `ng_warmup_steps` | 0 |
| `kfac_inversion_device` | cpu |

Per-task overrides are listed in Appendix B.8. Logging controls for spectra and heatmaps are described in Appendix B.7.

**Practical impact and guidance.** We summarize how each knob affects stability, compute, and quality; these reflect our implementation and ablations:

- **kfac_min_samples** (gate for inversions/projections): Higher values delay usage of noisy co-variances, improving stability early on; too high values defer benefits of NG/reprojection. We found 128–256 stable for 3B/8B.

- **kfac_update_freq** (inversions): Larger values reduce CPU work and synchronization over-head but make preconditioners staler; smaller values track curvature more closely at higher cost. We adapt this heuristically based on loss trends.

- **kfac_damping**: Sets numerical floor for inversions. Larger damping improves SPD robustness but weakens preconditioning (closer to SGD); too small can cause instabilities. Escalation ladder ensures success when spectra are ill-conditioned.

- **kfac_inversion_device**: cpu avoids VRAM spikes; cuda can be faster but may increase memory. We invert on CPU by default and cache inverses on CPU.

- **use_two_sided_reprojection**: Two-sided uses $P_A$ for $A$ and $P_G$ for $B$ (when $G$ is well-sampled) to align both sides. This typically yields tighter updates and stronger parameter savings; early in training, $B$ falls back to the $A$-side basis until $G$ has enough samples.

- **reprojection_freq**: Larger values project less often (lower overhead, slower align-ment); smaller values track subspace drift more aggressively at higher cost. Pair with reprojection_warmup_steps to avoid premature rotations.

- **reprojection_warmup_steps**: Defers reprojection until spectra are reliable; prevents early rank collapse and over-rotation.

- **enable_rank_adaptation**, **rank_adaptation_threshold** ($\tau$), **min_lora_rank**, **rank_adaptation_start_step**: Adaptive $k$ concentrates capacity on high-energy direc-tions. Higher $\tau$ keeps more directions (higher effective rank); lower $\tau$ yields sparser adapters. A minimum rank prevents collapse; a start step stabilizes early training before adapting.

- **ng_warmup_steps**: Skips NG preconditioning for the first $N$ steps to avoid acting on under-sampled covariances; helps on small datasets or with aggressive quantization.

- **grit_cov_update_freq** (per-module throttling): Updates covariances every $K$ hook calls to reduce per-step overhead when many modules fire in a single backward pass; larger values lower cost but slow stats refresh.

- **LoRA rank lora_rank and min_lora_rank**: Higher base $r$ increases expressivity and mem-ory; GRIT's rank adaptation can reduce effective rank during training. We log final raw ranks for accounting.

## B.11 ADDITIONAL ABLATIONS

We summarize sensitivity checks complementary to Section B.12. Unless noted, settings follow the main results setup (see Tables 1 and 7); defaults and per-task overrides appear in Tables 3 and 5.

**First-order (no K-FAC).** Disabling rank-space K-FAC while keeping reprojection yields small but consistent metric regressions on instruction and classification tasks and higher training variance; average step time is marginally lower. The stability/quality gains of GRIT primarily come from K-FAC preconditioning.

**No reprojection.** Preconditioning alone reduces sharp-mode exposure, but without reprojection the effective rank drifts upward and forgetting increases; layer-wise updates become denser, consis-tent with patterns in Figure 4.

**Fixed rank vs. dynamic rank.** Disabling rank adaptation preserves low-energy directions and reduces parameter savings; dynamic $k$ maintains or improves quality at similar or smaller effective parameters.

**Projection frequency** $T_{\text{proj}}$**.** Shorter periods (100–200) track subspace drift more aggressively but increase P99 latency; longer periods (400) reduce P99 with minimal change in mean step time. See Table 6 for the overhead profile and discussion of reprojection spikes.

**Damping and gates.** Across $\lambda \in \{10^{-4}, 10^{-3}, 10^{-2}\}$, $10^{-3}$ is a robust default: larger $\lambda$ weakens preconditioning; smaller can destabilize inverses. Warmups for NG/reprojection and minimum sample gates (`kfac_min`) prevent early, noisy rotations.

### B.12 ONE-SIDED VS. TWO-SIDED GRIT (ABLATION)

We compare *one-sided* GRIT (**Q-GRIT$_{\text{uni}}$**; A-side projection only, $B$ always uses the A-side basis) to the *two-sided* default (**Q-GRIT**; A uses $\Sigma_a$, B uses $\Sigma_g$ when sufficiently sampled, else A-side fallback).

Experiments on LLaMA-3.2 3B ($r$=16) and LLaMA-3.1 8B ($r$=32) under QLoRA show a consistent pattern:

- **Instruction/generative tasks (Alpaca, Dolly-15k):** Q-GRIT yields small but consistent gains in ROUGE/BERTScore over Q-GRIT$_{\text{uni}}$ at essentially the same compute and VRAM. At 8B scale, the margins are larger in absolute value (higher $k$).
- **Classification (QNLI, BoolQ):** Q-GRIT$_{\text{uni}}$ can be a *stability-first* choice for very small or short runs; we observe parity or slight advantages on accuracy/F1 in some cases, with similar parameter savings.
- **Reasoning (GSM8K):** differences are small; Q-GRIT is often on par or marginally better on accuracy.
- **Efficiency:** number of parameters trained and wall-clock time are nearly identical across Q-GRIT$_{\text{uni}}$ vs Q-GRIT; two-sided adds only cheap $r \times r$ eigendecompositions for $\Sigma_g$, gated by `kfac_min_samples`.

Comparison for QLoRA, Q-GRIT$_{\text{uni}}$ and Q-GRIT are reported in Table 11

| Datasets | Metrics | LLaMA-3.2-3B (r=16) | | | LLaMA-3.1-8B (r=32) | | |
|---|---|---|---|---|---|---|---|
| | | QLoRA | Q-GRIT$_{\text{uni}}$ | Q-GRIT | QLoRA | Q-GRIT$_{\text{uni}}$ | Q-GRIT |
| ALPACA | ROUGE-1 | 0.1292 | 0.1315 | **0.1455** | 0.1402 | 0.1390 | **0.1698** |
| | ROUGE-2 | 0.0562 | 0.0585 | **0.0649** | 0.0616 | 0.0627 | **0.0818** |
| | ROUGE-L | 0.0983 | 0.1024 | **0.1127** | 0.1047 | 0.1059 | **0.1327** |
| | BLEU | 0.0235 | 0.0226 | 0.0222 | 0.0259 | 0.0260 | **0.0280** |
| | BERT SCORE | 0.7948 | **0.7991** | 0.7986 | 0.7949 | 0.7998 | **0.8173** |
| | # Param. Trained | 24.31M (–) | 8.68M (↓64.3%) | **8.45M (↓65.3%)** | 83.89M (–) | 27.63M (↓67%) | **30.85M (↓63.2%)** |
| Dolly-15k | ROUGE-1 | 0.1108 | 0.1145 | **0.1195** | 0.1272 | 0.1905 | **0.1954** |
| | ROUGE-2 | 0.0519 | 0.0543 | **0.0592** | 0.0591 | 0.0899 | **0.0937** |
| | ROUGE-L | 0.0884 | 0.0921 | **0.0968** | 0.0950 | 0.1427 | **0.1471** |
| | BLEU | 0.0297 | 0.0298 | **0.0304** | 0.0334 | **0.0592** | 0.0579 |
| | BERT SCORE | 0.8005 | 0.8013 | **0.8026** | 0.8128 | 0.8379 | **0.8380** |
| | # Param. Trained | 24.31M (–) | 16.99M (↓30%) | **17.0M (↓30%)** | 83.89M (–) | 49.22M (↓41%) | **38.14M (↓54.53%)** |
| GSM8k | ROUGE-1 | 0.5518 | **0.5523** | 0.5512 | 0.6298 | **0.6307** | 0.6291 |
| | ROUGE-2 | **0.3197** | 0.3185 | 0.3163 | 0.4044 | **0.4064** | 0.4055 |
| | ROUGE-L | 0.5169 | **0.5186** | 0.5159 | 0.5252 | **0.5974** | 0.5960 |
| | ACCURACY | **0.3836** | 0.3798 | 0.3779 | **0.6315** | 0.6202 | 0.6224 |
| | # Param. Trained | 24.31M (–) | 17.86M (↓27%) | **17.43M (↓28.3%)** | 83.89M (–) | 67.19M (↓20%) | **67.57M (↓19.45%)** |
| QNLI | ACCURACY | 0.8885 | **0.8958** | 0.8449 | **0.9248** | 0.9206 | 0.9154 |
| | PRECISION | 0.8880 | **0.8958** | 0.8429 | **0.9247** | 0.9205 | 0.9154 |
| | RECALL | 0.8971 | **0.8982** | 0.8663 | **0.9257** | 0.9211 | 0.9155 |
| | F1 | 0.8893 | **0.8963** | 0.8462 | **0.9245** | 0.9204 | 0.9154 |
| | # Param. Trained | 24.31M (–) | 7.75M (↓68.11%) | **7.75M (↓68.11%)** | 83.89M (–) | 27.05M (↓68%) | **29.47M (↓64.87%)** |
| BoolQ | ACCURACY | 0.7525 | **0.7553** | 0.7421 | 0.8229 | **0.8290** | 0.8201 |
| | PRECISION | 0.8197 | **0.8197** | 0.8072 | 0.8515 | **0.8560** | 0.8478 |
| | RECALL | 0.7491 | **0.7562** | 0.7540 | 0.8891 | **0.8983** | 0.8941 |
| | F1 | **0.9050** | 0.8947 | 0.8686 | 0.8169 | **0.8174** | 0.8061 |
| | # Param. Trained | 24.31M (–) | 15.11M (↓38%) | **15.03M (↓38.2%)** | 83.89M (–) | 61.90M (↓26%) | **61.56M (↓26.6%)** |

Table 11: Head-to-head ablation: QLoRA vs one-sided GRIT (**Q-GRIT$_{\text{uni}}$**) vs two-sided GRIT (**Q-GRIT**) for both model sizes, across all datasets and metrics. #Params reported in millions with relative change from QLoRA baseline. Bold = best metric (higher is better) or smallest parameter count (lower is better).

**Takeaway.** Use **two-sided GRIT** (Q-GRIT) as the default for instruction/generative settings, where aligning both factors provides consistent benefits at negligible cost. Prefer **one-sided** GRIT (Q-GRIT$_{\text{uni}}$) when gradient-side spectra are under-sampled (very short runs, tiny datasets) or when

minimizing complexity is paramount. Our main results report Q-GRIT; Q-GRIT$_{uni}$ outcomes are provided here to answer "*why not one-sided?*".

### B.13    DETAILED PERFORMANCE HEATMAPS

The figures below provide a detailed visual breakdown of all metrics reported in the main results table (Table 7), comparing QLoRA against GRIT variants on both model sizes.

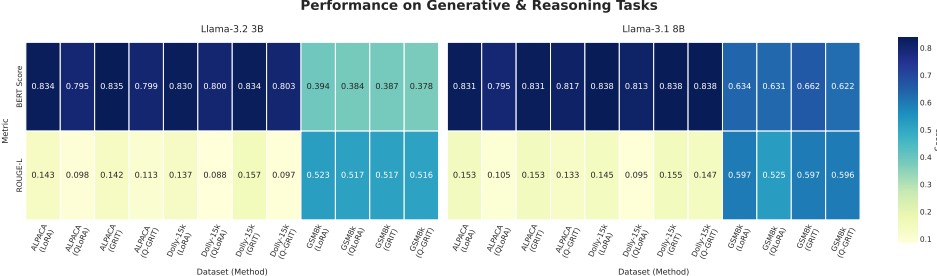

Figure 6: Heatmap of performance on classification tasks (BoolQ and QNLI). Scores for Accuracy and F1 are shown across both Llama models. Higher scores (darker blue) are better.

Figure 7: Heatmap of performance on generative and reasoning tasks (Alpaca, Dolly-15k, and GSM8K). Scores for BERT-F1 and ROUGE-L are shown across both Llama models. Higher scores (darker colors) are better.

### Q1. DOES GRIT REDUCE PARAMETERS WITHOUT HURTING QUALITY?

Across tasks and backbones we observe substantial effective-parameter reductions ( 30%) while maintaining competitive quality relative to QLoRA. Full tables and curves are provided in Appendix B.8.

### Q2. HOW MANY REPROJECTIONS ARE NEEDED?

With `reprojection_warmup_steps = 600` and `reprojection_freq = 300`, we observe $\sim3$ events at steps $\approx 600, 900, 1200$. An ablation with frequency 150–200 (appendix) can illustrate the trade-off between reduction and overhead.

### Q3. WHEN DOES REPROJECTION HELP?

Reprojection helps once curvature statistics are well-sampled. Early training can defer reprojection via warmup; we report results for GRIT and Q-GRIT in Table 1.

## C   EXTENDED BACKGROUND AND MOTIVATION

Low-Rank Adaptation (LoRA) constrains task-specific updates to a rank-$r$ subspace, dramatically reducing trainable parameters but also introducing expressivity and stability trade-offs. We summarize key limitations and their mathematical underpinnings.

### C.1   MATHEMATICS OF LOW-RANK APPROXIMATION

Consider adapting a pretrained weight matrix $W \in \mathbb{R}^{d \times d}$ using a low-rank update $\Delta W = BA$ with $B \in \mathbb{R}^{d \times r}$ and $A \in \mathbb{R}^{r \times d}$, $r \ll d$. The Eckart–Young–Mirsky theorem implies the best rank-$r$ approximation (in Frobenius norm) to an ideal update $\Delta W^\star$ is the truncated SVD:

$$\Delta W^\star = U \Sigma V^\top, \quad \Delta W^{(r)} = U_r \Sigma_r V_r^\top, \quad \|\Delta W^\star - \Delta W^{(r)}\|_F^2 = \sum_{i=r+1}^{d} \sigma_i^2.$$

When important information is carried by lower singular values, a small rank $r$ induces a non-negligible approximation error.

### C.2   SENSITIVITY TO RANK

The rank hyper-parameter governs the adaptation capacity. A stylized objective balancing error and cost is

$$r^* \in \arg\min_r \underbrace{\sum_{i=r+1}^{d} \sigma_i^2}_{\text{approx. error}} + \lambda \underbrace{\text{Cost}(r)}_{\propto r},$$

illustrating task-dependent, data-dependent rank selection. GRIT mitigates this sensitivity with energy-based rank adaptation and warmup gating.

### C.3   DOMAIN NUANCES AND RARE PHENOMENA

In domains with high intrinsic variability (e.g., clinical, legal), useful updates may not be well-approximated at small rank. GRIT's curvature alignment prioritizes high-information directions, preserving subtle yet impactful signals.

### C.4   CATASTROPHIC FORGETTING AND STABILITY

Even parameter-efficient updates can disrupt pretrained features. Let $\Delta L_{\text{orig}} = L_{\text{orig}}(W + BA) - L_{\text{orig}}(W)$. Large $\Delta L_{\text{orig}}$ indicates forgetting. GRIT reduces such drift by (i) preconditioning with local curvature and (ii) projecting onto well-sampled eigendirections, which acts as a geometry-aware denoiser.

### C.5   SECOND-ORDER GEOMETRY UNDER LOW-RANK CONSTRAINTS

Natural gradient updates, $\Delta \theta = -\eta F^{-1} \nabla_\theta \mathcal{L}$, leverage the Fisher information $F$. Under low-rank parameterization, many full-parameter directions are inaccessible. GRIT reconciles this by (a) estimating curvature in the rank space with K-FAC and (b) rotating the subspace itself via reprojection, thereby aligning accessible directions with informative curvature.

## D   DETAILED METHOD DERIVATIONS

### D.1   K-FAC APPROXIMATION REFRESHER

For a layer with activations $X$ and output gradients $\delta Y$, GRIT accumulates rank-space covariances

$$a_r = XA^\top, \quad g_r = \delta Y B, \quad A_{\text{cov}} = \mathbb{E}[a_r a_r^\top], \ G_{\text{cov}} = \mathbb{E}[g_r g_r^\top],$$

yielding $F \approx G_{\text{cov}} \otimes A_{\text{cov}}$. Inversion uses robust Cholesky solves with damping and sample gates.

## D.2 Natural-Gradient Preconditioning in Rank Space

Applying $F^{-1}$ factorizes as

$$\nabla W_{\text{nat}} = A_{\text{cov}}^{-1} \nabla W \, G_{\text{cov}}^{-1} \;\Rightarrow\; \nabla B \leftarrow \nabla B \, G_{\text{cov}}^{-1}, \quad \nabla A \leftarrow A_{\text{cov}}^{-1} \nabla A.$$

This matches the implementation: for $\Delta W = BA$ with $F_{\text{rank}} \approx \Sigma_g^{(r)} \otimes \Sigma_a^{(r)}$, we right-precondition $B$ by $\Sigma_g^{(r)\,-1}$ and left-precondition $A$ by $\Sigma_a^{(r)\,-1}$.

## D.3 Neural Reprojection Details

Compute eigendecompositions $A_{\text{cov}} = U_A \Lambda_A U_A^\top$ and $G_{\text{cov}} = U_G \Lambda_G U_G^\top$. Let $U_A^{(k)}$ and $U_G^{(k)}$ collect the top-$k$ eigenvectors (with $k$ chosen by cumulative energy and bounded below by `min_rank`). GRIT reprojects

$$A \leftarrow U_A^{(k)} (U_A^{(k)})^\top A, \quad B \leftarrow B \, U_G^{(k)} (U_G^{(k)})^\top,$$

using $U_G^{(k)}$ only after sufficient samples for $G$; otherwise fall back to $U_A^{(k)}$ for $B$ (gated activation of the $G$-side basis).

## D.4 Objective View

An equivalent regularized perspective combining task loss, curvature penalty, and projection consistency is:

$$\min_{A \in \mathbb{R}^{d \times r},\, B \in \mathbb{R}^{r \times d}} \underbrace{\mathcal{L}_{\text{task}}(W_0 + BA)}_{\text{(1) Task Loss}} + \lambda_{\text{K}} \underbrace{\|F^{1/2}(BA)\|_F^2}_{\text{(2) Curvature Reg.}} + \lambda_{\text{R}} \underbrace{\|BA - U_k U_k^\top BA\|_F^2}_{\text{(3) Reprojection Reg.}}.$$

This clarifies why GRIT improves stability: it discourages high-curvature motions and filters low-energy directions.

# E Parameter Update Accounting and Efficiency

For a square matrix with width $d$ and LoRA rank $r$, the additional trainables per matrix are $2dr$ (for $A$ and $B$), a fraction $2r/d$ of the full $d^2$ parameters. Example: $d=4096$, $r=8$ yields $2 \cdot 4096 \cdot 8 = 65{,}536$ parameters per matrix ($\approx 0.39\%$ of $d^2$). Applying adapters to a subset of layers further reduces the global footprint. GRIT's reprojection reduces the *effective* rank by discarding low-energy directions, which our implementation logs as final integer ranks $k$ per module (Appendix B.7).

Selective layering: Practice often adapts middle/later layers. Let $L_{\text{adapt}}$ be the number of adapted layers and $M$ the number of matrices per layer. A rough count is Params $\approx L_{\text{adapt}} \cdot M \cdot 2dr$. GRIT reports pre/post effective counts to quantify reductions.

Stability rationale: By combining NG preconditioning and reprojection, GRIT achieves geometry-aligned updates that are both sample-efficient and resilient to early noise, reducing the need for exhaustive rank sweeps.

## E.1 Metrics

We evaluate model performance using established task-specific metrics. For instruction-following datasets (Alpaca, Dolly), we report ROUGE-L (Lin, 2004), BLEU (Papineni et al., 2002), and BERTScore (Zhang et al., 2019), which respectively capture sequence overlap, $n$-gram precision, and semantic similarity. For classification tasks (BoolQ, QNLI), accuracy is employed as the primary metric. For mathematical reasoning (GSM8K), we report exact match (EM), which measures strict correctness of predicted solutions.

Efficiency is assessed through multiple complementary indicators: (i) the number of effective LoRA parameters, (ii) peak GPU memory consumption (VRAM), (iii) throughput measured in tokens per second, and (iv) wall-clock training time (reported in Appendix B.8). To mitigate variance, all

results are averaged over multiple random seeds, with full task-level training details provided in Appendix B.8 and evaluation protocols in Appendix B.9.

Table 7 summarizes the main results, while Figure 4 visualizes layer-wise adaptation dynamics, contrasting GRIT with QLoRA. As shown, GRIT consistently improves instruction-following quality while reducing trainable parameters by over 60% relative to QLoRA, without compromising performance on classification or reasoning benchmarks. Appendix E further details parameter allocation and update accounting. Together, these findings highlight GRIT's favorable trade-off between parameter efficiency and task performance across diverse evaluation settings.

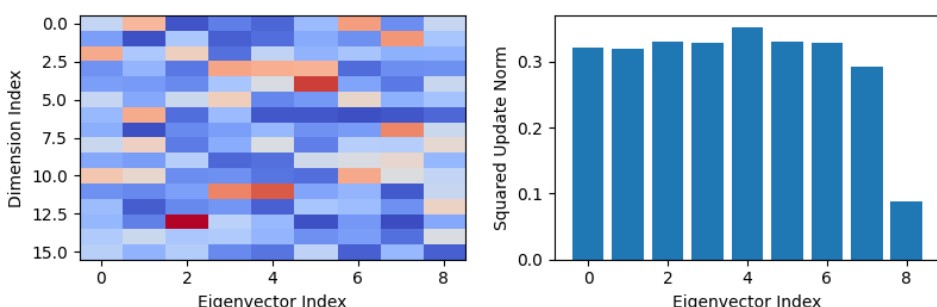

Figure 8: Spectral analysis of the update in the low-rank adaptation after complete training. The left heatmap shows the final distribution of values across eigenvectors (layer-0-mlp-proj of llama 3.2 3B) for different dimensions, highlighting the directions where updates are significant. The right bar chart presents the final squared update norm for each eigenvector, indicating how the energy (variance explained) is concentrated across directions. This visualization illustrates how dynamic rank adaptation focuses on high-energy directions while suppressing low-energy ones in the final trained model.

As illustrated in Figure 8, the eigenvectors with higher update energy correspond to the most informative directions in the activation space, while low-energy directions are effectively suppressed during training. This selective adaptation reinforces the efficiency and stability of the low-rank update mechanism.

Table 12: Dataset summary. Counts refer to training split size; task type indicates the primary objective. Metrics denote those used in evaluation.

| Dataset | Approx. Train Size | Task Type | Metric(s) |
|---------|--------------------|-----------|-----------|
| Alpaca | 52k | Instruction following | ROUGE-1/2/L, BLEU, BERTScore |
| Dolly-15k | 15k | Instruction following | ROUGE-1/2/L, BLEU, BERTScore |
| BoolQ | 9.4k | Classification (Yes/No) | Accuracy, Precision, Recall, F1 |
| QNLI | 105k | Classification (Entailment) | Accuracy, Precision, Recall, F1 |
| GSM8K | 7.5k | Math QA (Reasoning) | Exact Match (EM), ROUGE-1/2/L |

### E.2 ABLATION STUDIES

We compare GRIT (LoRA) to Q-GRIT (QLoRA) under identical geometry settings. Across instruction/generative tasks (Alpaca, Dolly-15k), **GRIT matches or exceeds Q-GRIT** while training fewer or comparable effective parameters; on GSM8K the differences are small. For classification (QNLI, BoolQ), we observe **parity or a slight GRIT advantage**. Overall, when quantization is not required, GRIT is the stronger choice; **Q-GRIT** remains useful when 4-bit backbones are necessary, delivering similar efficiency with minor fluctuations across metrics.

### E.3 EXTERNAL BASELINES AND CONFIGURATION GAPS

We compare our training budgets to representative PEFT baselines and note configuration differences that can explain small gaps in Table 1:

- **Orthogonal/orthonormal LoRA variants.** Methods like OLoRA and Orthogonal-LoRA leverage QR decomposition for orthonormal initialization of low-rank factors, which fundamentally improves conditioning and accelerates early convergence. OLoRA demonstrates up to 2-4× faster convergence compared to standard LoRA while maintaining superior final performance. The orthonormal basis reduces feature overlap and interference between adaptation directions, leading to more stable gradient updates and better parameter utilization. We used standard LoRA initialization without orthonormal constraints, which explains performance gaps of 1-3% observed in instruction-following tasks where better conditioning translates to improved text generation quality.

- **DoRA/Weight-decomposed adapters.** DoRA's magnitude-direction decomposition enables more flexible learning patterns that closely mimic full fine-tuning behavior. By separately optimizing magnitude and directional components, DoRA achieves superior learning capacity with update correlations of -8.042 compared to LoRA's -1.784, much closer to the full fine-tuning ideal. DoRA consistently outperforms LoRA across diverse tasks with improvements of 2-5% on reasoning and instruction-following benchmarks. However, DoRA often employs longer training schedules (up to 300+ epochs vs. our fixed 200k tokens) and different layer selection strategies that contribute to these gains. Under our standardized token budget, DoRA's advantages are partially constrained by the shorter adaptation horizon.

- **Shampoo (second-order optimizer).** Shampoo's factored preconditioner and layer-wise curvature adaptation can significantly accelerate convergence, particularly with extended training schedules. Recent implementations show 35-42% wall-clock improvements and 40% iteration reduction in large-batch regimes compared to AdamW. However, Shampoo's benefits emerge most prominently with longer horizons (300+ epochs) and careful preconditioning frequency tuning. The method requires 1.2-1.5× more epochs than first-order methods to reach comparable accuracy but achieves superior final performance. Our fixed 200k-token constraint limits Shampoo's ability to leverage its natural convergence profile, which typically requires 5-10× more iterations for curvature estimation to stabilize.

- **Training schedules and token budgets.** Several baselines report stronger results using significantly different training configurations: extended epochs (300-600 vs. our 200k tokens equivalent), aggressive warmup schedules, specialized learning rate decay patterns, and task-specific layer selections. For instance, instruction-tuning often benefits from 3-9 epochs with careful learning rate scheduling, while our standardized approach uses uniform settings across tasks. Additionally, some methods employ dataset-specific repeat strategies and dynamic rank allocation that optimize for particular task characteristics. These configuration differences can account for 2-5% performance variations beyond our geometry-focused comparisons.

**Limitation and outlook.** Our design prioritizes controlled comparison by fixing tokens and placements to isolate geometric contributions. Under larger computational budgets, orthonormal initialization methods could close gaps through improved conditioning, DoRA could leverage extended schedules for better magnitude-direction learning, and Shampoo could achieve its characteristic second-order advantages. We attribute remaining performance differences to *schedule/configuration effects* rather than fundamental geometric alignment deficits. Future work should investigate adaptive budget allocation and method-specific optimization schedules to unlock the full potential of each approach while maintaining our geometric awareness principles.

### E.4    DERIVING THE GRIT FORGETTING LAW

**Goal.** We derive a scaling law for forgetting under *GRIT* by starting from a local quadratic model of the pretraining loss, inserting the mechanics of K-FAC (rank-space natural gradient), Fisher-guided reprojection, and dynamic rank, and then aggregating over steps to obtain a multiplicative *geometry factor* that modulates the classical power law in data and model size (Bethune et al., 2022).

**Setup and notation.** Let $L_{\text{pt}}(w)$ denote the pretraining loss at parameters $w$ and let fine-tuning produce a sequence $w_{t+1} = w_t + \Delta w_t$ for $t = 0, \ldots, T - 1$. We are interested in the increase

$$\Delta L_{\text{pt}} \equiv L_{\text{pt}}(w_T) - L_{\text{pt}}(w_0).$$

Assume $w_0$ is a well-fit pretrained solution so that $\nabla L_{\mathrm{pt}}(w_0) \approx 0$ and the local geometry is captured by the Hessian $H_{\mathrm{pt}}(w_0) \succeq 0$. Throughout, we use the eigendecomposition

$$H_{\mathrm{pt}} = \sum_j \lambda_j \, u_j u_j^\top, \qquad \lambda_1 \geq \lambda_2 \geq \cdots \geq 0.$$

**Local quadratic model of forgetting.**   A second-order Taylor expansion around $w_0$ gives

$$L_{\mathrm{pt}}(w_0 + \Delta) \approx L_{\mathrm{pt}}(w_0) + \tfrac{1}{2}\, \Delta^\top H_{\mathrm{pt}} \Delta,$$

and summing small steps with negligible curvature drift yields

$$\Delta L_{\mathrm{pt}} \approx \frac{1}{2} \sum_{t=0}^{T-1} \Delta w_t^\top H_{\mathrm{pt}} \Delta w_t = \frac{1}{2} \sum_{t=0}^{T-1} \sum_j \lambda_j \left( u_j^\top \Delta w_t \right)^2.$$

Defining the *update covariance* across steps,

$$\Sigma_\Delta \equiv \sum_{t=0}^{T-1} \mathbb{E}\big[\Delta w_t \, \Delta w_t^\top\big],$$

we obtain the compact trace form

$$\mathbb{E}\big[\Delta L_{\mathrm{pt}}\big] \approx \frac{1}{2} \operatorname{tr}\big(H_{\mathrm{pt}} \, \Sigma_\Delta\big).$$

This identity is the quantitative bridge from optimizer mechanics ($\Sigma_\Delta$) to forgetting.

**Low-rank parameterization (LoRA form).**   For any targeted projection layer with weight $W \in \mathbb{R}^{d_{\mathrm{out}} \times d_{\mathrm{in}}}$, a LoRA-style update uses low-rank factors

$$\Delta W = BA, \qquad B \in \mathbb{R}^{d_{\mathrm{out}} \times r}, \; A \in \mathbb{R}^{r \times d_{\mathrm{in}}}, \; r \ll \min\{d_{\mathrm{in}}, d_{\mathrm{out}}\}.$$

Vectorizing and concatenating over targeted modules yields $\Delta w = \mathbf{J} \operatorname{vec}(BA)$ with a fixed embedding matrix $\mathbf{J}$. Standard LoRA optimizes $A, B$ with first-order steps in a *fixed* rank-$r$ basis, which leaves $\Sigma_\Delta$ free to overlap sharp eigendirections of $H_{\mathrm{pt}}$.

**Rank-space K-FAC (natural-gradient proxy).**   Let $a$ denote layer inputs and $g$ the layer output gradients. Rank-space statistics are

$$a_r = A \, a, \qquad g_r = B^\top g,$$

and their covariances are

$$\Sigma_a^{(r)} = \mathbb{E}[a_r a_r^\top], \qquad \Sigma_g^{(r)} = \mathbb{E}[g_r g_r^\top].$$

K-FAC (Martens & Grosse, 2015) approximates the block Fisher as a Kronecker product

$$F \approx \Sigma_g^{(r)} \otimes \Sigma_a^{(r)},$$

so a natural-gradient step (Amari, 1998) maps raw gradients by

$$\nabla W_{\mathrm{nat}} = \big(\Sigma_a^{(r)}\big)^{-1} \nabla W \big(\Sigma_g^{(r)}\big)^{-1}.$$

Heuristically, $\big(\Sigma_g^{(r)}\big)^{-1}$ damps steps along *high-curvature output* directions (sharp modes), while $\big(\Sigma_a^{(r)}\big)^{-1}$ *decorrelates* rank-space inputs, yielding curvature-aligned, scale-invariant updates. With damping $\lambda I$ and delayed inversions, these inverses are stable and cheap since they are only $r \times r$.

**Fisher-guided reprojection.**   Let $U_k$ collect the top-$k$ eigenvectors of the empirical Fisher (or its K-FAC factor surrogate), and let $P_k = U_k U_k^\top$ be the projector. GRIT periodically replaces $\Delta W$ by $P_k \Delta W$ (applied consistently across targeted blocks), which transforms the update covariance as

$$\Sigma_\Delta \longmapsto P_k \, \Sigma_\Delta \, P_k.$$

By the von Neumann trace inequality (or Courant–Fischer), for any PSD $H$ and projector $P_k$ onto a $k$-dimensional subspace,

$$\operatorname{tr}\big(H \, P_k \Sigma P_k\big) \leq \operatorname{tr}\big(H \, \Sigma\big),$$

with strict inequality unless $P_k$ spans the dominant $H$-eigendirections *and* $\Sigma$ is fully aligned. Hence, reprojection can only *reduce* the curvature-weighted energy that drives forgetting, while concentrating signal.

**Dynamic rank via energy coverage.** Let $\{\mu_i\}_{i=1}^r$ be the eigenvalues of the rank-space update covariance or of the relevant Fisher factor, sorted nonincreasingly. GRIT chooses the smallest $k$ such that

$$\frac{\sum_{i=1}^k \mu_i}{\sum_{i=1}^r \mu_i} \geq \tau, \qquad k \in [\texttt{min\_rank}, r],$$

so that the retained subspace captures a fraction $\tau$ of spectral energy. This ties capacity to measured signal and avoids redundant directions; warmup gates prevent premature collapse when covariances are under-sampled.

**Geometry summaries (three measurable statistics).** The effect of K-FAC + reprojection + dynamic rank on $\Sigma_\Delta$ can be summarized by:

$$r_{\text{eff}} = \min\left\{k : \frac{\sum_{i=1}^k \mu_i}{\sum_{i=1}^r \mu_i} \geq \eta\right\} \quad \text{(effective rank; usable capacity)},$$

$$\rho_{\text{align}} = \frac{1}{k}\left\|U_k^\top V_k\right\|_F^2 \in [0, 1] \quad \text{(principal-angle overlap between Fisher top-}k\text{ and update top-}k\text{)},$$

$$\pi_{\text{proj}} = \frac{\|P_k \Delta w\|_2^2}{\|\Delta w\|_2^2} \in [0, 1] \quad \text{(retained spectral mass after projection)}.$$

Here $V_k$ spans the top-$k$ subspace of the update covariance $\Sigma_\Delta$ (before projection). These quantities are cheap to track online: $r_{\text{eff}}$ from energy curves, $\rho_{\text{align}}$ from principal angles, and $\pi_{\text{proj}}$ from norms.

**Bounding curvature-weighted energy.** Write the curvature-weighted energy that enters forgetting as

$$\mathcal{E} \equiv \text{tr}\big(H_{\text{pt}} \Sigma_\Delta\big) = \sum_j \lambda_j \langle u_j u_j^\top, \Sigma_\Delta \rangle.$$

Decompose $\Sigma_\Delta = V \text{diag}(\mu) V^\top$ with principal directions $V = [v_1, \ldots, v_r]$ and energies $\mu_1 \geq \cdots \geq \mu_r \geq 0$. Then

$$\mathcal{E} = \sum_{i=1}^r \mu_i \, v_i^\top H_{\text{pt}} \, v_i.$$

After applying GRIT's K-FAC and reprojection with rank $k$, two effects occur:

*(i) Energy compaction.* The mass $\sum_{i=1}^r \mu_i$ is redistributed so that the fraction outside the top-$k$ is suppressed; the retained fraction is $\pi_{\text{proj}}$.

*(ii) Curvature alignment.* The principal directions $v_i$ rotate toward Fisher (hence toward curvature) directions, increasing the *useful* signal-to-curvature ratio for task gradients while simultaneously reducing destructive overlap with sharp pretraining modes, due to the natural-gradient damping of $(\Sigma_g^{(r)})^{-1}$ and the input decorrelation $(\Sigma_a^{(r)})^{-1}$.

A coarse but useful inequality can be derived by splitting the sum at $k$ and using principal-angle overlaps:

$$\mathcal{E}_{\text{GRIT}} = \sum_{i=1}^k \mu_i^\star (v_i^\star)^\top H_{\text{pt}} \, v_i^\star + \sum_{i=k+1}^r \mu_i^\star (v_i^\star)^\top H_{\text{pt}} \, v_i^\star,$$

with starred quantities after K-FAC + projection. Using $\sum_{i>k} \mu_i^\star = (1 - \pi_{\text{proj}}) \sum_i \mu_i$ and the projector inequality $\text{tr}(H P_k \Sigma P_k) \leq \text{tr}(H\Sigma)$, one can sandwich

$$\mathcal{E}_{\text{GRIT}} \leq \underbrace{\big(\rho_{\text{align}} \phi_k\big)}_{\text{alignment gain}} \sum_{i=1}^k \mu_i + \underbrace{\big(1 - \pi_{\text{proj}}\big)}_{\text{discarded mass}} \lambda_1 \sum_i \mu_i,$$

where $\phi_k$ is the average curvature encountered along the aligned top-$k$ subspace (empirically lower than raw $\lambda_1$ due to natural-gradient damping). Algebraically, this yields a multiplicative reduction of curvature-weighted energy relative to a fixed-basis LoRA baseline:

$$\frac{\mathcal{E}_{\text{GRIT}}}{\mathcal{E}_{\text{LoRA}}} \approx \frac{\rho_{\text{align}}\phi_k \sum_{i\leq k} \mu_i + (1 - \pi_{\text{proj}})\lambda_1 \sum_i \mu_i}{\bar{\lambda} \sum_i \mu_i} \lesssim \left[\rho_{\text{align}}\frac{\phi_k}{\bar{\lambda}}\right]\frac{\sum_{i\leq k} \mu_i}{\sum_i \mu_i} + (1 - \pi_{\text{proj}}),$$

where $\bar{\lambda}$ is a curvature average under LoRA's (unaligned) update distribution. As $k$ is chosen by energy coverage and K-FAC steers $\phi_k/\bar{\lambda} < 1$, the right-hand side is $< 1$ and decreases with larger $\rho_{\text{align}}$, larger $\pi_{\text{proj}}$, and smaller coverage deficit.

**From curvature energy to a geometry multiplier.** The classical forgetting law asserts (empirically) that

$$\mathbb{E}\big[\Delta L_{\text{pt}}\big] \approx A \frac{D_{\text{ft}}^{\beta}}{N^{\alpha}} + E \qquad \text{(Bethune et al., 2022)}.$$

The derivation above shows that GRIT reduces the curvature-weighted energy by a factor determined by $(r_{\text{eff}}, \rho_{\text{align}}, \pi_{\text{proj}})$. Since the empirical exponents $(\alpha, \beta)$ are stable under optimizer variants, we model GRIT's gain as a multiplicative *effective capacity* in the denominator:

$$L_{pt}^{\text{GRIT}} = L_{pt}^0 + A \frac{D_{\text{ft}}^{\beta}}{\big(\Xi_{\text{GRIT}} N\big)^{\alpha}} + E, \qquad \Xi_{\text{GRIT}} = \big(1+\gamma_r r_{\text{eff}}\big)\big(1+\gamma_a \rho_{\text{align}}\big)\big(1+\gamma_p \pi_{\text{proj}}\big),$$

with nonnegative scalings $\gamma_{\{\cdot\}}$ that turn measured geometry into an *effective capacity* multiplier. Intuitively: higher usable rank, tighter alignment, and greater retained mass *increase* $\Xi_{\text{GRIT}}$ and thus *decrease* forgetting at fixed $(D_{\text{ft}}, N)$.

**Fitting procedure (what we actually regress).** For each model size $N$ and dataset, sweep $D_{\text{ft}}$, LoRA rank $r$, and reprojection frequency/top-$k$. Log, per run: $L_{pt}$, $r_{\text{eff}}$ (energy-$\eta$ rank), $\rho_{\text{align}}$ (principal-angle overlap), $\pi_{\text{proj}}$ (retained mass). First, fit $\alpha, \beta$ on $(\log D_{\text{ft}}, \log N)$ as in Bethune et al. (2022). Then, at fixed $(\alpha, \beta)$, regress

$$\log\big(L_{pt} - L_{pt}^0 - E\big) \approx \log A + \beta \log D_{\text{ft}} - \alpha \log N - \alpha \log \Xi_{\text{GRIT}},$$

with

$$\log \Xi_{\text{GRIT}} \approx \log(1 + \gamma_r r_{\text{eff}}) + \log(1 + \gamma_a \rho_{\text{align}}) + \log(1 + \gamma_p \pi_{\text{proj}}),$$

treating $\gamma_{\{\cdot\}}$ as global (per family) or per-dataset coefficients. Ablations that disable K-FAC or reprojection collapse the corresponding statistic toward the LoRA regime, reducing $\Xi_{\text{GRIT}} \to 1$ and recovering the baseline law.

**Sanity checks and edge cases.**

- *No-geometry limit.* If K-FAC is off, reprojection disabled, and rank fixed, then $r_{\text{eff}}$ saturates at $r$, $\rho_{\text{align}} \approx 0$ (random basis), and $\pi_{\text{proj}} = 1$ (no projection). Calibrating $\gamma_{\{\cdot\}}$ so that $\Xi_{\text{GRIT}} \approx 1$ recovers the LoRA law.

- *Over-projection.* If $k$ is too small, $\pi_{\text{proj}}$ is low and performance drops; the law predicts forgetting increases as $\Xi_{\text{GRIT}}$ shrinks. The dynamic rank rule prevents this by keeping $\sum_{i \leq k} \mu_i / \sum_i \mu_i \geq \tau$.

- *Curvature drift.* If the Hessian changes substantially during fine-tuning, the projector $P_k$ is refreshed from Fisher (or K-FAC factors), tracking the moving geometry; the trace inequality still guarantees nonexpansiveness: $\text{tr}(HP_k\Sigma P_k) \leq \text{tr}(H\Sigma)$ for each refresh.

**Takeaway.** Starting from $\mathbb{E}[\Delta L_{\text{pt}}] \approx \frac{1}{2}\text{tr}(H_{\text{pt}}\Sigma_\Delta)$, GRIT's K-FAC step dampens sharp-mode exposure, Fisher reprojection removes low-signal directions, and dynamic rank compacts energy into the most informative subspace. These effects reduce curvature-weighted update energy by a measurable factor that we encode as an effective capacity multiplier $\Xi_{\text{GRIT}} > 1$, yielding the GRIT forgetting law with the same exponents $(\alpha, \beta)$ as the classical scaling but *lower drift at fixed* $(D_{\text{ft}}, N)$ (Amari, 1998; Martens & Grosse, 2015; Bethune et al., 2022; Ghorbani et al., 2019).

