# OpenReview forum: "GRIT: Geometry-Aware PEFT with K-FAC Preconditioning, Fisher-Guided Reprojection, and Dynamic Rank Adaptation"
_ICLR.cc/2026/Conference — ICLR 2026 Conference Withdrawn Submission_

### Official Review · Reviewer_zX3u · 2025-10-27

**Soundness:** 2
**Presentation:** 3
**Contribution:** 2
**Rating:** 4
**Confidence:** 3

**Summary:**

This paper proposes GRIT, a geometry-aware PEFT method that augments LoRA with three components: K-FAC preconditioning, Fisher-guided reprojection and dynamic rank adaptation. The goal is to keep or slightly improve task performance while substantially reducing trainable parameters

**Strengths:**

### Strengths
- The three components are well-motivated and can be slotted into standard LoRA/QLoRA pipelines with modest overhead.
- The proposed methods empirically achieve comparable or better performance with fewer trainable parameters; analyses visualize how updates concentrate in high-signal directions.
- The paper explains the intended role of each component in the algorithm, providing a good understanding of the method.

**Weaknesses:**

- Motivation vs. measurement (catastrophic forgetting): The paper’s motivation centers on reducing forgetting via controlling tail mass and effective rank, but the experimental section largely reports task performance (and some geometric proxies) without direct forgetting metrics. Relying on a scaling-law narrative alone leaves the central claim under-evaluated. A convincing test would report pre-train–capability drift and compare GRIT to LoRA/QLoRA and other baselines under matched budgets.
- Ablations are not sufficient for causality: GRIT is a three-part method, yet the paper mainly shows how reprojection/preconditioning change update density. What’s missing is a performance & forgetting ablation across the three components of the method. Without this, the necessity and contribution of each component remain unsupported.

**Questions:**

- Section 1.2 suggests mass above $ \tau$ mainly affects pretrain loss. Why does the contribution exhibit a threshold effect rather than a smooth Hessian-weighted sum across directions? What determines $\tau$ empirically or theoretically?
- Also in section1.2, the claim is that spreading updates across more directions raises the chance of overlapping with sharp pretrain Hessian modes, hence more forgetting. But if fine-tune and pretrain Hessians are weakly correlated, why is constraining effective rank sufficient to reduce pre-train loss drift?

---

### Official Review · Reviewer_TMDx · 2025-10-27

**Soundness:** 2
**Presentation:** 1
**Contribution:** 3
**Rating:** 4
**Confidence:** 4

**Summary:**

The authors present GRIT, a geometry-aware parameter-efficient fine-tuning (PEFT) framework for large language models (LLMs) that augments the canonical LoRA/QLoRA adapter schemes with three key innovations: (1) natural gradient preconditioning in the low-rank adapter space using K-FAC approximations to account for local curvature; (2) periodic reprojection of adapters onto the dominant eigendirections of the Fisher information matrix to align updates with high-signal, low-interference directions; and (3) dynamic rank adaptation that allocates PEFT capacity in response to the evolving Fisher spectrum. Experiments across five benchmarks on LLaMA backbones systematically demonstrate that GRIT achieves comparable or better performance than existing PEFT baselines, while substantially reducing the number of trained parameters.

**Strengths:**

1. Clear Motivations: The manuscript provides a compelling diagnosis of the limitations of geometry-agnostic PEFT methods, articulating both intuitively (Section 1.2) and mathematically (quadratic loss expansion w.r.t. Hessian eigenspectrum, Eq. for $\Delta L_{\mathrm{pt}}$) why local curvature and alignment matter.

2. Innovative Algorithmic Components: Following these motivations, GRIT introduces a cohesive combination: K-FAC natural gradient preconditioning (Section 2.2)—well-motivated for the PEFT/LoRA context; Fisher-guided neural reprojection (Section 2.3), which dynamically aligns the low-rank subspace; and spectrum-driven rank adaptation (Section 2.4), moving beyond fixed, arbitrary rank choices.

3. Strong Use of Visuals: Figures such as Fig. 1 (loss decomposition), Fig. 2 (pipeline diagram), and Fig. 3 (PCA of parameter updates) crystallize the geometric thrust and effectiveness of GRIT compared to LoRA. Fig. 4 (ablation of updated parameters across layers) gives tangible insight into sparsity gains and task-specific allocations, while Fig. 5 (heatmap of dynamic rank across layers/modules) supports the claims of spectrum-adaptive capacity.

**Weaknesses:**

1. Unclear Computational and Memory Overheads: While the proposed method introduces new mechanisms to adjust the update process in LoRA, the practical computational and memory costs remain unclear. For instance, in Line 210, the authors suggest adjusting the gradient direction using the Fisher Information Matrix (FIM). However, the additional time and memory overhead introduced by this operation are not discussed. Since PEFT methods generally emphasize the memory cost, it is necessary to provide those data.

2. Presentation Issue – Excessive Use of Emphasis Formatting: The manuscript frequently employs italic and bold text, sometimes even in combination. This inconsistent and excessive use of emphasis makes the paper appear visually cluttered and distracts from the main content. It is recommended that the authors minimize such formatting to maintain a cleaner, more formal, and professional presentation style.

3. Limited Empirical Evaluation: The paper presents relatively limited empirical evaluations, focusing primarily on tuning LLaMA 3.2. This narrow experimental scope weakens the overall persuasiveness of the results. The reviewer recommends that the authors include a more comprehensive set of experiments and comparisons, especially with models and benchmarks commonly used in previous related studies for fair comparison. Such additions would help substantiate the claimed advantages and improve the paper’s overall credibility.

4. Lack of Ablation Studies: The paper introduces several contributing factors, such as the adjustment of gradient directions, the use of adaptive ranks, and other design choices. However, no ablation studies are provided to isolate and evaluate the individual impact of each component. Including systematic ablation experiments would greatly strengthen the paper by clarifying which factors contribute most significantly to the overall performance improvements.

**Questions:**

1. On Fisher vs. Hessian Alignment: Given the recent evidence (e.g., Dauphin et al., 2024) that key sharpness/interference components may not be captured by the Fisher, would GRIT’s performance change if the projection or curvature alignment used the full Hessian or higher-order approximations? How practical or necessary is this—any experiments or theory to support the choice?

2. Sensitivity to Spectral Hyperparameters: How robust is GRIT to variations in $\tau$ (energy threshold), reprojection frequency $T_{\text{proj}}$, and damping factor $\lambda$? Please provide ablations quantifying the effects of under- or over-estimating these inputs on both parameter count and performance.

3. [Key Issue] Runtime/Overhead at Larger Scale/Longer Horizon: The paper claims only modest training overhead; the reviewer would surprised to see this since GRIT involves multiple extra computation. Moreover, do these results hold for larger LMs or over much longer training scripts (e.g., continued pretraining or multitask setting)?

4. [Key Issue] Batch size issue and variance: since for large models, we typically use a relatively small batch size. This renders a large variance to the gradient in practice. How would this affect some of the steps in GRIT, such as the preconditioning?

5. Practical Limitations and Stability Risks: Following Q4, the discussion of algorithmic stability and early-phase Fisher noise (in Section 2.3, Table 8) is helpful, but mostly documented as limitations and mitigations, rather than explored through targeted ablations or robustness analysis. For real-world deployments, the risks of under- or over-rotating the subspace (projection sensitivity), misestimating the desired energy threshold for rank selection ($\tau$ in Eq. for $k$), or catastrophic forgetting from update misalignment are flagged but insufficiently quantified or simulated. There is a lack of systematic ablation on the sensitivity of GRIT to these hyperparameters and schedules.

6. Curvature Law Empirical Grounding Needs Strengthening: The proposed scaling law for forgetting (Section 3.1) is stated in theoretical terms and supported with qualitative reasoning, but lacks detailed quantitative fits or visualizations linking changes in each geometry term ($r_\text{eff}$, $\rho_\text{align}$, $\pi_\text{proj}$) to downstream loss or interference. This undercuts its practical utility and falsifiability

---

### Official Review · Reviewer_UJQd · 2025-10-31

**Soundness:** 3
**Presentation:** 3
**Contribution:** 3
**Rating:** 4
**Confidence:** 2

**Summary:**

This paper proposes GRIT, a geometrically aware parametric fine-tuning (PEFT) framework, which addresses the geometrically independent limitations of methods such as LoRA through K-FAC preprocessing, Fisher-guided reprojection, and dynamic rank adaptation. GRIT, while maintaining LoRA parameterization, utilizes natural gradient proxy to optimize direction, suppress drift, and concentrate effective capacity. In the instruction following, understanding, and inference benchmarks of the LLaMA series models, it reduces the average trainable parameters by approximately 46% (25%-80% between tasks). And its performance is on par with or exceeds that of LoRA/QLoRA. Its forgetting law conforms to the curvature modulation power law, and its drift is significantly lower than that of traditional methods. Moreover, in the comparison with strong baselines such as Orthogonal-LoRA and IA3, it occupies the forefront of parameter update - performance retention. This achievement breaks through the trade-off between learning and forgetting in PEFT, providing a new paradigm for the efficient adaptation of large-scale language models.

**Strengths:**

1. The core value of this paper lies in reconstructing the parameter efficient fine-tuning design logic with geometric perception, breaking through the limitations of traditional methods.

2. The reproducibility statement is sufficient. However, its placement may not be appropriate. The content preceding the References section already spans 14 pages, which seems excessively long.

**Weaknesses:**

1. The originality of some technical components is limited, which weakens the breakthrough value of the key idea. For instance, techniques such as K-FAC curvature approximation, Fisher information matrix-guided optimization, and dynamic rank allocation are not proposed for the first time in this paper.

2.The readability of the paper could be further improved.

**Questions:**

1. The paper proposes that geometric perception is a key breakthrough of PEFT, but it does not clearly define the essential difference between GRIT's geometric perception and other PEFT methods involving curvature, such as LoRA variants based on Hessian approximations. For instance, some works have also attempted to use simplified Hessian to guide the update direction of LoRA. How does GRIT's K-FAC approximation differ from these methods in terms of curvature estimation accuracy, computational cost, and adaptability?

2. The abstract is not concise enough, and the symbols in the formulas are not explained. For example, what does $D_{ft}^\beta$ mean?
In addition, it is uncommon to include formulas in an abstract and highlight them with boxes. I personally suspect these were directly generated by an LLM.

3. The overall structure and readability of the paper need improvement, especially in sect. 1, where the background of PEFT is not clearly introduced, and the comparison with related work is not sufficiently detailed.

4. Some definitions are difficult to understand. For example, in line 110, what is H_{pt}? What does P represent?

---

### Official Review · Reviewer_Kzmp · 2025-11-02

**Soundness:** 1
**Presentation:** 1
**Contribution:** 1
**Rating:** 0
**Confidence:** 3

**Summary:**

This paper is incredibly hard to read, and I suspect this was fully generated with an LLM.

I can't make sense of the math, the experiments don't make sense (ROUGE-1 score for GSM-8K?), and the writing is all over the place.

I refuse to provide a review.

**Strengths:**

I believe this is LLM-generated slop.

**Weaknesses:**

I believe this is LLM-generated slop.

**Questions:**

I believe this is LLM-generated slop.

**Details Of Ethics Concerns:**

I believe this is LLM-generated slop.

---

### Note · Authors · 2025-12-03

I have read and agree with the venue's withdrawal policy on behalf of myself and my co-authors.